# IOSTOM: Offline Imitation Learning from Observations Via State Transition Occupancy Matching

**Quang Anh Pham, Janaka Chathuranga Brahmanage, Tien Mai, Akshat Kumar**
Singapore Management University
{qa.pham.2025, janakat.2022}@phdcs.smu.edu.sg
{atmai, akshatkumar}@smu.edu.sg

## Abstract

Offline Learning from Observation (LfO) focuses on enabling agents to imitate expert behavior using datasets that contain only expert state trajectories and separate transition data with suboptimal actions. This setting is both practical and critical in real-world scenarios where direct environment interaction or access to expert action labels is costly, risky, or infeasible. Most existing LfO methods attempt to solve this problem through state or state-action occupancy matching. They typically rely on pretraining a discriminator to differentiate between expert and non-expert states, which could introduce errors and instability—especially when the discriminator is poorly trained. While recent discriminator-free methods have emerged, they generally require substantially more data, limiting their practicality in low-data regimes. In this paper, we propose IOSTOM (*Imitation from Observation via State Transition Occupancy Matching*), a novel offline LfO algorithm designed to overcome these limitations. Our approach formulates a learning objective based on the joint state visitation distribution. A key distinction of IOSTOM is that it first excludes actions entirely from the training objective. Instead, we learn an *implicit policy* that models transition probabilities between states, resulting in a more compact and stable optimization problem. To recover the expert policy, we introduce an efficient action inference mechanism that *avoids training an inverse dynamics model*. Extensive empirical evaluations across diverse offline LfO benchmarks show that IOSTOM substantially outperforms state-of-the-art methods, demonstrating both improved performance and data efficiency.

## 1 Introduction

Imitation learning is a framework in machine learning where agents learn to perform tasks by mimicking expert demonstrations rather than learning through trial-and-error or explicit reward signals [33, 37, 15]. This approach is particularly useful in environments where designing reward functions is difficult or costly. Its practical relevance spans a wide range of domains, including robotics, healthcare, and autonomous driving, where expert behavior is available but reinforcement learning is either too risky, time-consuming, or expensive to deploy [35, 46, 24]. By leveraging expert demonstrations, imitation learning enables faster deployment of intelligent systems and facilitates safer exploration in complex, real-world environments.

A variant of this framework, known as Imitation from Observations or Learning from Observations (LfO), focuses on learning policies using only state trajectories without access to the expert's actions. This setting presents unique challenges, such as inferring intent and disambiguating optimal behavior from partial information, but also broadens applicability to scenarios where action data is unavailable or hard to record. For example, in video-based learning from human demonstrations in household

tasks (e.g., cleaning or cooking), it is often infeasible to capture the precise motor commands or control actions, making observation-only learning a practical and valuable approach.

Recent developments in imitation learning from observations have increasingly focused on scenarios where a limited set of expert state-only trajectories is complemented by sub-optimal state-action demonstrations. While this setup has practical appeal, many existing methods rely on distribution-matching frameworks that operate over complex input tuples such as $(s, a, s')$ or $(s, s', s'')$, where $s, s'$ represents a state and $a$ an action [18, 38]. These formulations appear to be sample-inefficient due to the structural complexity of the inputs. Furthermore, some approaches require estimating a discriminator to support training [27, 48], which can be unreliable in low-data or high-dimensional settings [39]. Other methods rely on learning an inverse dynamics model to recover unobserved expert actions, which introduces approximation errors that may degrade the quality of the learned policy [43, 50]. *To the best of our knowledge, no existing method in the LfO setting addresses all of these limitations simultaneously.*

We aim to address the aforementioned limitations in this work. Our central idea is to ignore sub-optimal actions and instead focus on learning the *transition probabilities between consecutive states*, leading to a simple and compact learning objective that only involves joint state pairs $(s, s')$. We then develop an efficient method to recover the expert policy without requiring an inverse dynamics model. Specifically, our contributions are as follows:

(i) By first disregarding sub-optimal actions in the demonstration data, we propose to learn *state-to-state transition probabilities*, which can be interpreted as an *implicit policy* that encapsulates the actual state-action policy. We then formulate the learning problem as matching *joint state visitation distributions* and leverage convexity and Lagrangian duality to derive a tractable joint-state Q-learning procedure. This training formulation, in addition to being *discriminator-free*, is *significantly simpler and more compact* than prior approaches that rely on action annotations, as it only involves consecutive state pairs $(s, s')$.

(ii) We further introduce two novel strategies for efficiently extracting a policy from the learned Q-function. First, we propose a Q-weighted behavior cloning (BC) approach, which is theoretically equivalent to the standard advantage-weighted BC but offers a more compact and stable formulation. Second, we propose a single-stage process for recovering the expert policy without estimating an *inverse dynamics model*, thereby avoiding approximation errors that could degrade policy quality.

(iii) We validate our LfO framework using state-of-the-art benchmarks, demonstrating that our algorithm, IOSTOM, significantly outperforms existing methods. The implementation of IOSTOM is publicly available at `https://github.com/quanganh1999/IOSTOM`.

## 2 Related Work

**Learning from Observations** Different from Learning from Demonstrations (LfD) [37, 35] using expert state-action dataset, Learning from Observations (LfO) [45] addresses the challenge of imitation learning when expert actions are unavailable, relying instead on state-only expert trajectories. LfO research can be broadly distinguished into online and offline paradigms. In online LfO setting, the agent can actively interact with the environment [44, 49]. Recent advancement in online LfO focuses on improving adversarial imitation learning (AIL) approaches [14, 6]. The core idea of AIL relies on generative adversarial networks (GANs) [11] where a generator policy learns to imitate an expert, while a discriminator differentiates between agent-generated and expert data. In addition to online LfO, its offline setting has also received significant interest due to practical constraints of many real-world scenarios, where continuous interaction is costly or risky. It assumes access to state-only expert demonstrations and an action-labeled background dataset from other interactions [50]. A common approach trains an inverse dynamics model (IDM) on background data to infer expert actions, then applies Behavior Cloning (BC) [43, 4]. However, BC needs extensive, high-quality expert data and can suffer from compounding errors, exacerbated by IDM inaccuracies [34]. Another line adapts the Distribution Correction Estimation (DICE) framework [28]. These methods (e.g., PW-DICE [48], SMODICE [27], LobsDICE [18]) use a discriminator to estimate density ratios as pseudorewards for downstream RL. While avoiding an explicit IDM, their success depends on discriminator quality and RL robustness. Recently, DILO [38] bypass both IDM and discriminator learning by solving the dual

of an occupancy matching objective, directly optimizing a utility function. This function, measuring long-term divergence from expert visitation, is used to extract the imitation policy.

**Imitation Learning via Distribution Matching:**  Distribution Matching objective is a powerful tool in Reinforcement Learning (RL) that has demonstrated its effectiveness in exploration [25], goal-conditioned RL [26, 1], and especially Imitation Learning (IL). Many popular IL methods such as BC, GAIL [13], and DAgger [36] can be formulated as statistical divergence minimization problems [10]. This minimization can be performed over the state, state-action, or trajectory space, resulting in different IL approaches [30]. The well-known DICE-family algorithms [21, 19, 18, 27] also optimizes state or state-action visitation distribution matching problems between the learner and expert via their dual formulations [29]. They often require learning a discriminator to estimate the log-ratio for distribution correction. Recently, [39] introduce ReCOIL, a discriminator-free method that also optimizes the duality of the state-action occupancy matching problem. This work is closely related to our IOSTOM, as both learn a score function that assigns high values to expert samples and low values to non-expert samples. However, IOSTOM focuses on solving the state-transition occupancy matching problem instead of the standard state-action one to address the LfO problem. Our setting is generally considered more challenging than the LfD setting targeted by ReCOIL [17], mainly due to the absence of expert actions in LfO.

## 3   Background

**Markov Decision Process.**  We consider a Markov Decision Process (MDP) defined by the tuple $\mathcal{M} = \langle \mathcal{S}, \mathcal{A}, \mathcal{T}, \mathcal{R}, p_0, \gamma \rangle$, where $\mathcal{S}$ denotes the set of states, $\mathcal{A}$ set of actions, $p_0$ represents the distribution of initial states, $\mathcal{R} : S \times A \to \mathbb{R}$ defines the reward function for each state-action pair, and $\mathcal{T} : S \times A \to S$ is the transition function, i.e., $\mathcal{T}(s'|s,a)$ is the probability of reaching state $s' \in \mathcal{S}$ when action $a \in \mathcal{A}$ is taken at state $s \in \mathcal{S}$. The parameter $\gamma \in [0,1)$ is the discount factor. In reinforcement learning (RL), the goal is to find a policy that maximizes the expected long-term accumulated reward, i.e., $\max_\pi \left\{ \mathbb{E}_{(s,a) \sim d^\pi}[\mathcal{R}(s,a)] \right\}$, where $d^\pi(s,a)$ is the occupancy measure (or state-action visitation distribution) of policy $\pi$. The definitions of $d^\pi(s,a)$ and other common visitation distributions are include in Table 1.

| | State Distribution | State-Action Distribution | Joint Distribution | Transition Distribution |
|---|---|---|---|---|
| Notation | $d^\pi(s)$ | $d^\pi(s,a)$ | $d^\pi(s,a,s')$ | $d^\pi(s,s')$ |
| Support | $\mathcal{S}$ | $\mathcal{S} \times \mathcal{A}$ | $\mathcal{S} \times \mathcal{A} \times \mathcal{S}$ | $\mathcal{S} \times \mathcal{S}$ |
| Definition | $(1-\gamma)\sum_{t=1}^\infty \gamma^t P(s_t = s \mid \pi)$ | $d^\pi(s)\pi(a\|s)$ | $d^\pi(s,a)\mathcal{T}(s'\|s,a)$ | $\sum_{\mathcal{A}} d^\pi(s,a,s')$ |

Table 1: Overview on different stationary distributions adapted from [50]

**Offline Imitation Learning from Observations.**  Different from standard Imitation Learning, Learning from observations (LfO) relaxes the requirement of action in expert dataset. In offline LfO setting, we assume access to an expert observation-only dataset $D_E = \{s, s'\}$ and a suboptimal interaction dataset $D_I = \{s, a, s'\}$. We also denote the respective visitation distributions of the expert and suboptimal datasets $D_E$ and $D_I$ as $d^E$ and $d^I$. Several methods have been proposed to handle this challenging scenario. For instance, SMODICE [27], a state-of-the-art approach for learning from observations (LfO), minimizes an upper bound of KL-divergence $\mathbb{D}_{\mathrm{KL}}[d^\pi(s) \,\|\, d^E(s)]$ via the objective:

$$\min \mathbb{E}_{d^\pi(s)} \left[ \log \frac{d^I(s)}{d^E(s)} \right] + \mathbb{D}_f \left[ d^\pi(s,a) \,\|\, d^I(s,a) \right].$$

where $\mathbb{D}_f$ denotes an $f$-divergence between two distributions. LobsDICE [18] proposes a similar formulation:

$$\min \mathbb{D}_{\mathrm{KL}} \left( d^\pi(s,s') \,\|\, d^E(s,s') \right) + \alpha \, \mathbb{D}_{\mathrm{KL}} \left( d^\pi(s,a) \,\|\, d^I(s,a) \right),$$

DILO [38] introduces a discriminator-free approach via solving another objective:

$$\min \mathbb{D}_f \left[ \beta \, d^\pi(s,s',a') + (1-\beta) \, d^I(s,s',a'), \; \beta \, d^E(s,s',a') + (1-\beta) \, d^I(s,s',a') \right],$$

where $d^\pi(s,s',a') = d^\pi(s,s')\pi(a'|s')$.

Most methods (except DILO) rely on a learned discriminator to predict distribution correction ratios based on $s$ or $(s, s')$, which can be unreliable in low-data or high-dimensional settings [39]. Although DILO is discriminator-free, it requires costly triplet samples $(s, s', s'')$ and a non-standard visitation structure, which limits its sample efficiency. In contrast, our proposed method **IOSTOM** is discriminator-free and only involves joint state pairs $(s, s')$ during learning. This leads to a more compact representation and improved sample efficiency over prior approaches. Furthermore, IOSTOM is the only approach directly minimizing $\mathbb{D}_f[d^\pi(s, s') \,\|\, d^E(s, s')]$ which is the main objective of LfO [44, 18].

# 4 IOSTOM - Imitation Learning via State Transition Occupancy Matching

We present a novel framework for imitation learning from observations, which is structured into two sequential stages. The first stage focuses on recovering the state-transition probabilities, denoted as $g(s' \mid s)$, which represents the probability of transitioning to the next state $s'$ given the current state $s$. In the second stage, we recover a policy based on the learned state transition model $g(s' \mid s)$. The key insight behind our method is to simplify the LfO problem by initially ignoring the action information in the dataset. By doing so, we treat the state transition model $g(s' \mid s)$ as a form of "*implicit policy*" that governs the behavior of the demonstrator. This abstraction allows us to bypass the need for explicit action labels during the early phase of learning.

## 4.1 Joint State Q-learning Formulation

Our approach centers on recovering the transition probability between states, denoted as $g(s' \mid s)$, which can be viewed as an *implicit policy*. This transition model can be computed based on the underlying policy and environment dynamics as: $g(s' \mid s) = \sum_a \pi(a \mid s)\mathcal{T}(s' \mid s, a)$. To facilitate training, we define the joint visitation distribution over state pairs $(s, s')$ as:

$$d^g(s, s') = d^\pi(s)g(s'|s) = \sum_a d^\pi(s, a)\mathcal{T}(s' \mid s, a),$$

where $d^\pi(s, a)$ is the state-action visitation distribution under policy $\pi$, recursively computed via the single-step transpose Bellman equation [29]:

$$d^\pi(s, a) = (1 - \gamma)d_0(s)\pi(a \mid s) + \gamma \sum_{s', a'} d^\pi(s', a')\mathcal{T}(s \mid s', a')\pi(a \mid s),$$

with $d_0(s)$ denoting the initial state distribution. To support learning from observations, our goal is to remove the dependence on actions in the visitation distribution. We introduce the following proposition to this end:

**Proposition 1.** *The joint visitation distribution $d^g(s, s')$ can be expressed as:*

$$d^g(s, s') = (1 - \gamma)d_0(s)g(s' \mid s) + \gamma g(s' \mid s) \sum_{\bar{s}} d^g(\bar{s}, s) \tag{1}$$

We note that the flow equality in Equation (1) depends solely on the joint state visitation distribution $d^g(s, s')$ and the state transition function $g(s' \mid s)$—the two key quantities we aim to recover.

Using the flow constraints described above and following the approach in [39], our main objective is to minimize the divergence between two joint state visitation distributions: $d^I_{\text{mix}}(s, s')$ and $d^{E,I}_{\text{mix}}(s, s')$, defined as follows:

$$d^I_{\text{mix}}(s, s') = \alpha d(s, s') + (1 - \alpha)d^I(s, s'), \text{ and } d^{E,I}_{\text{mix}}(s, s') = \alpha d^E(s, s') + (1 - \alpha)d^I(s, s'),$$

where $\alpha \in (0, 1)$ is a mixing hyperparameter. Here, $d^I_{\text{mix}}(s, s')$ represents a mixed visitation distribution combining the learned state-transition behavior with that of the suboptimal dataset, while $d^{E,I}_{\text{mix}}(s, s')$ represents a mixed distribution combining expert behavior and suboptimal behavior. Combining this with the flow constraints in (1), we formulate our learning problem as follows:

$$\max_{g, d \geq 0} \quad -\mathbb{D}_f(d^I_{mix}(s, s') \| d^{E,I}_{mix}(s, s'))$$

$$s.t. \quad d(s, s') = (1 - \gamma)d_0(s)g(s'|s) + \gamma g(s'|s) \sum_{\bar{s}} d(\bar{s}, s) \tag{2}$$

The constrained problem described above is convex in $d$ when the transition function $g$ is fixed. Following a similar approach to that in [39], the maximization over $d$ can be equivalently reformulated as an unconstrained optimization problem via Lagrangian duality. We formalize this result in the following proposition, with the full derivation provided in the appendix.

**Proposition 2.** *The constrained optimization problem in Equation* (2) *is equivalent to the following unconstrained max-min problem:*

$$\max_g \min_{Q(s,s')} \left\{ \alpha(1-\gamma)\, \mathbb{E}_{(s,s')\sim d_0}\left[Q(s,s')\right] + \mathbb{E}_{(s,s')\sim d_{mix}^{E,I}}\left[f^*\left(\gamma\mathbb{E}_{s''\sim g(\cdot|s')}\left[Q(s',s'')\right] - Q(s,s')\right)\right] \right.$$
$$\left. - (1-\alpha)\,\mathbb{E}_{(s,s')\sim d^I}\left[\gamma\mathbb{E}_{s''\sim g(\cdot|s')}\left[Q(s',s'')\right] - Q(s,s')\right] \right\}, \tag{3}$$

*where $Q(s,s')$ are the Lagrange multipliers and $f^*$ denotes the convex conjugate of a chosen convex function $f$.*

For notational convenience, we define $Q(s,g) = \mathbb{E}_{s'\sim g(\cdot|s)}\left[Q(s,s')\right]$. Using this shorthand, we write the objective function in (3) as (See Appendix B.3 for complete derivation):

$$\max_g \min_{Q(s,s')} \left\{ L(Q,g) = \alpha(1-\gamma)\,\mathbb{E}_{(s,s')\sim d_0}\left[Q(s,s')\right] + \alpha\mathbb{E}_{(s,s')\sim d^E}\left[f^*\left(\gamma Q(s',g) - Q(s,s')\right)\right] \right.$$
$$\left. + (1-\alpha)\,\mathbb{E}_{(s,s')\sim d^I}\left[\widetilde{f^*}\left(\gamma Q(s',g) - Q(s,s')\right)\right] \right\}, \tag{4}$$

where $\widetilde{f^*}(t) = f^*(t) - t$. This formulation learns a *joint-state value function* $Q(s,s')$, where actions are entirely ignored. While being more compact and manageable than prior formulations that rely on $(s,a,s')$ or even $(s,s',s'')$ tuples, our approach benefits from ignoring suboptimal actions in the data. This design helps mitigate the imbalance in offline datasets, where expert demonstrations lack action labels, whereas suboptimal trajectories contain fully observed actions.

## 4.2 Extreme V-learning

Solving the maximin objective in (4) can be done via dual optimization by alternating between optimizing $Q(s,s')$ and $g$. Specifically, we minimize $L(Q,g)$ over $Q$, and then maximize it over $g$ with $Q$ fixed. Following [47], the maximization over $g$ can be approximated by computing $\max_g Q(s,g)$ for each state $s$, which requires sampling from $g$. In offline RL, this is challenging due to out-of-distribution (OOD) issues when querying $Q$ on unseen state transitions [23]. To address this, we adopt the in-sample soft estimation from [9], replacing the hard maximization with a KL-regularized soft value: $V_Q^g(s) = \mathbb{E}_{s'\sim g(\cdot|s)}[Q(s,s') - \beta\log\frac{g(s'|s)}{\mu(s'|s)}]$, where $\mu$ is the behavior policy and $\beta$ controls the KL strength, keeping $g$ close to $\mu$ to avoid OOD samples. This is supported by the following proposition:

**Proposition 3.** $\max_g\{V_Q^g(s)\}$ *can be approximated via the following Extreme-V objective:*

$$\min_V \left\{ J(V\mid Q) = \mathbb{E}_{(s,s')\sim d_{mix}^{E,I}}\left[\exp(\omega(s,s')) + \omega(s,s') - 1\right] \right\}. \tag{5}$$

*where $\omega(s,s') = (Q(s,s') - V(s))/\beta$.*

Using this estimate, the $Q$-learning objective becomes:

$$\min_Q L(Q,V) = \alpha(1-\gamma)\,\mathbb{E}_{(s,s')\sim d_0}\left[Q(s,s')\right] + \alpha\mathbb{E}_{(s,s')\sim d^E}\left[f^*\left(\gamma V(s') - Q(s,s')\right)\right]$$
$$+ (1-\alpha)\mathbb{E}_{(s,s')\sim d^I}\left[\widetilde{f^*}\left(\gamma V(s') - Q(s,s')\right)\right]. \tag{6}$$

We optimize $Q$ and $V$ jointly: $Q$ by minimizing $L(Q,V)$, and $V$ by minimizing $J(V\mid Q)$. Crucially, $L(Q,V)$ is concave in $Q$, and $J(V\mid Q)$ is convex in $V$, forming a bi-concave/convex structure that ensures stable and convergent optimization.

**Proposition 4.** *Under any convex function $f$, $L(Q,V)$ is convex in $Q$, and the Extreme-V loss $J(V\mid Q)$ is convex in $V$.*

## 4.3 Policy Extraction

Typically, once the $Q$ and $V$ functions have been learned through Q-learning, a policy can be recovered using advantage-weighted behavior cloning (AW-BC) [31]. In our context, where we operate with an implicit policy $g(s' \mid s)$, the policy can be recovered by solving the following optimization:

$$\max_g \ \mathbb{E}_{(s,s')\sim d^I_{\text{mix}}(s,s')} \left[\exp\left(\tau(Q(s,s')-V(s))\right)\log g(s' \mid s)\right], \tag{7}$$

Here, $\tau > 0$ is a parameter controlling the sharpness of advantage weighting. We use $\mathcal{D}^I$ instead of the mixed dataset $\mathcal{D}^{E,I}_{\text{mix}}$ in (7), as the policy $\pi$ is extracted solely from $\mathcal{D}^I$. This is sufficient under the common coverage assumption that $d^I(s,s') > 0$ whenever $d^E(s,s') > 0$, as adopted in prior works [19, 18, 27].

A limitation of objective (7) is that the value function $V(s)$ in our setting is only an approximation obtained via the extreme-V surrogate, which can introduce noise and bias in the computation of advantages. To address this, we propose an alternative approach based solely on the $Q$-function. The following proposition shows that this alternative objective can, in theory, recover the same optimal implicit policy as the original advantage-weighted BC formulation.

**Proposition 5.** *The following Q-weighted behavior cloning objective returns the same optimal implicit policy as the original advantage-weighted BC formulation:*

$$\max_g \ \mathbb{E}_{(s,s')\sim d^I} \left[\exp\left(\tau Q(s,s')\right)\log g(s' \mid s)\right]. \tag{8}$$

Given the learned implicit transition model $g(s' \mid s)$, existing approaches often recover the expert policy $\pi(a|s)$ by training an inverse dynamics model (IDM), denoted as $\mathcal{I}(a \mid s, s')$. This is typically done by optimizing the following objective: $\max_{\mathcal{I}} \ \mathbb{E}_{(s,a,s')\sim d^I} \left[\log \mathcal{I}(a \mid s, s')\right]$, and then defining the recovered policy as $\pi(a \mid s) = \sum_{s'} \mathcal{I}(a \mid s, s')g(s' \mid s)$. While intuitive, this two-step approach has several limitations. First, decoupling the learning of $g(s' \mid s)$ and the recovery of $\pi(a \mid s)$ via a separate inverse model introduces additional sources of bias. Second, training the inverse dynamics model $\mathcal{I}(a \mid s, s')$ typically requires a significant amount of high-quality data. When the offline dataset $d^I$ contains a large proportion of low-quality or suboptimal data, the inverse model may be inaccurate—resulting in compounding approximation errors, as also noted in learning-from-observation (LfO) literature [18].

To address limitations of IDM-based recovery, we propose a single-stage policy extraction method that avoids training an inverse dynamics model. Our approach leverages the identity $g(s' \mid s) = \sum_a \mathcal{T}(s' \mid s, a)\pi(a \mid s)$. Using this, we rewrite the $Q$-weighted BC objective (8) as a direct optimization over $\pi$:

$$\max_\pi \ F(\pi) = \mathbb{E}_{(s,s')\sim d^I} \left[\exp\left(\tau Q(s,s')\right)\log\left(\sum_a \mathcal{T}(s' \mid s, a)\pi(a \mid s)\right)\right].$$

The objective $F(\pi)$, however, involves a log-sum over actions, making it difficult to optimize directly. We develop a tractable lower bound on this objective, which resembles a weighted behavior cloning loss over $\log \pi(a \mid s)$.

**Proposition 6.** *The objective $F(\pi)$ is lower-bounded by the following surrogate function $\widetilde{F}(\pi)$, up to an additive constant:* $\widetilde{F}(\pi) = \mathbb{E}_{(s,s')\sim\mathcal{D}^I} \left[\exp\left(\tau Q(s,s')\right)\sum_a \mathcal{I}(a \mid s, s')\log \pi(a \mid s)\right].$

While $\widetilde{F}(\pi)$ is a lower bound of the original objective $F(\pi)$, maximizing this surrogate function still promotes the maximization of $F(\pi)$ in practice. The primary advantage of the surrogate objective $\widetilde{F}(\pi)$ is that it contains the term $\sum_a \mathcal{I}(a \mid s, s')\pi(a \mid s)$, which can be empirically approximated using offline samples, thus avoiding the need to learn the inverse dynamics. In particular, we can empirically approximate $\widetilde{F}(\pi)$ as:

$$\widetilde{F}(\pi) \approx \mathbb{E}_{(s,a,s')\sim d^I} \left[\exp\left(\tau Q(s,s')\right)\log \pi(a \mid s)\right],$$

where the expectation is taken over offline trajectories $(s, a, s')$. We note that a similar weighted behavior cloning formulation was used in [38], although without providing theoretical justification. Empirically, their results demonstrate that this single-stage approach can outperform the traditional two-step method involving inverse dynamics modeling.

## 5 Practical Algorithm

The common choices of $f$-divergence function in the literature can be KL or Pearson $\chi^2$. In IOSTOM, we choose the $\chi^2$ divergence function with its convex conjugate function $f^*(x) = \frac{x^2}{4} + x$. Our objective (6) becomes (complete derivation can be found in the Appendix B.8):

$$\min_Q L(Q, V) = (1 - \gamma) \mathbb{E}_{d_0(s,s')} Q(s, s') + \gamma \mathbb{E}_{s \sim d^E}[V(s)] - \mathbb{E}_{s,s' \sim d^E}[Q(s, s')]$$
$$+ \frac{1}{4\alpha} \mathbb{E}_{s,s' \sim d_{mix}^{E,I}}[(\gamma V(s') - Q(s, s'))^2].$$

The $\min_Q -\alpha \mathbb{E}_{(s,s') \sim \mathcal{D}^E}[Q(s, s')]$ term in the above objective which effectively encourages maximizing the $Q$-values of expert transitions can lead to *unbounded* growth in $Q$, potentially resulting in learning instability. To address this issue, we adopt a technique from [2] that constrains the expert $Q$-values, and propose the following practical Q-learning objective (with derivation in Appendix B.9):

$$\widetilde{L}(Q, V) = (1 - \gamma) \mathbb{E}_{d_0}[Q(s, s')] + \frac{1 - \alpha}{4\alpha} \mathbb{E}_{d^I}\left[(\gamma V(s') - Q(s, s'))^2\right]$$
$$+ \frac{1}{4} \mathbb{E}_{d^E}\left[\left(Q(s, s') - \frac{2}{1 - \gamma}\right)^2\right]. \tag{9}$$

Finally, to estimate the term $\mathbb{E}_{(s,s') \sim d_0}[Q(s, s')]$, we sample $(s, s')$ pairs uniformly from the offline dataset rather than from a policy rollout. This empirical estimation, adopted in prior works [8, 38], helps reduce overfitting and improves the robustness of the learned policy by leveraging a diverse range of initial transitions. We present main steps of our IOSTOM algorithm in Algorithm 1.

## 6 Experiments

In this section, we compare IOSTOM with previous state-of-the-art approaches on diverse sets of environments and tasks from the D4RL benchmark [7], and real world data. Particularly, we aim to answer the following main questions: **(Q1)** Can IOSTOM outperform other baselines on standard LfO benchmarks? (Section 6.1) **(Q2)** Is our algorithm still robust with limited expert data? (Section 6.2) **(Q3)** How well IOSTOM perform when learning from experts of different dynamics? (Section 6.3) **(Q4)** What is the performance of IOSTOM on real-world instances (Section 6.4)? We also provide implementation details and additional experiments in the Appendix C.

---

**Algorithm 1** IOSTOM

1: **Input:** Expert dataset $D^E$, suboptimal dataset $D^I$
2: Initialize Q, V functions and policy networks $Q_\phi, V_\omega, \pi_\theta$
3: Set target network parameters $\phi' \leftarrow \phi$
4: **for** $t = 1, 2, \cdots, N$ **do**
5:     Sample mini-batches from $D^E$ and $D^I$
6:     *# Update V using $J(V, Q)$ in Equation* (5)
7:     $\omega \leftarrow \omega - \eta \nabla_\omega \widetilde{J}(V_\omega | Q_{\phi'})$
8:     *# Update Q using $\widetilde{L}(Q, V)$ in Equation* (9)
9:     $\phi \leftarrow \phi - \eta \nabla_\phi L(Q_\phi, V_\omega)$
10:     *# Update policy via weighted BC*
11:     $w(s, s') \leftarrow \exp(\tau Q_{\phi'}(s, s'))$
12:     $\theta \leftarrow \theta + \eta \nabla_\theta \mathbb{E}_{(s,a,s') \sim d^I}[w(s, s') \log \pi_\theta(a \mid s)]$
13:     *# Update target network*
14:     $\phi' \leftarrow \lambda \phi + (1 - \lambda)\phi'$
15: **end for**
16: **Output:** Imitation policy $\pi_\theta$

---

**Baselines and experimental setup** We choose three SOTA LfO methods in the literature as our main baselines: SMODICE [27], PW-DICE [48], and DILO [38]. Both SMODICE and PW-DICE require learning a discriminator. The main difference between them is that SMODICE aims to minize the KL-divergence distance of state visitation distributions between learner and expert while PW-DICE uses Wasserstein distance [16] instead. DILO is the recent SOTA discriminator-free method for LfO. We train all algorithms for 1 million gradient steps with 5 random seeds and monitor the *normalized score* $= 100 * \frac{\text{method score - random score}}{\text{expert score - random score}}$ [7] during training. The average normalized score of last 10 evaluations is used to assess the performance of different methods.

### 6.1 Offline IL from Observations

To answer the question **(Q1)**, we use the same offline LfO benchmark from DILO [39] with datasets constructed from the D4RL framework [7]. Specifically, we evaluate methods on 8 Mujoco envi-

| Suboptimal Dataset | Env | LfD approaches | | | LfO approaches | | | | Expert |
|---|---|---|---|---|---|---|---|---|---|
| | | BC (expert data) | BC (full dataset) | ReCOIL | SMODICE | PW-DICE | DILO | IOSTOM | |
| random+ expert | hopper | $4.52_{\pm1.42}$ | $5.64_{\pm4.83}$ | $108.18_{\pm3.28}$ | $106.56_{\pm0.53}$ | $108.09_{\pm2.39}$ | $86.35_{\pm38.00}$ | $\mathbf{109.32}_{\pm1.08}$ | 111.33 |
| | halfcheetah | $2.2_{\pm0.01}$ | $2.25_{\pm0.00}$ | $80.20_{\pm6.61}$ | $85.55_{\pm1.39}$ | $86.11_{\pm4.39}$ | $91.53_{\pm0.27}$ | $\mathbf{93.02}_{\pm0.40}$ | 88.83 |
| | walker2d | $0.86_{\pm0.61}$ | $0.91_{\pm0.5}$ | $102.16_{\pm7.19}$ | $107.93_{\pm1.26}$ | $107.48_{\pm0.53}$ | $\mathbf{108.31}_{\pm0.18}$ | $107.98_{\pm0.20}$ | 106.92 |
| | ant | $5.17_{\pm5.43}$ | $30.66_{\pm1.35}$ | $126.74_{\pm4.63}$ | $126.08_{\pm0.73}$ | $\mathbf{126.89}_{\pm1.17}$ | $125.39_{\pm2.37}$ | $\mathbf{128.19}_{\pm1.52}$ | 130.75 |
| random+ few-expert | hopper | $4.84_{\pm3.83}$ | $3.0_{\pm0.54}$ | $97.85_{\pm17.89}$ | $58.30_{\pm9.96}$ | $75.04_{\pm14.21}$ | $104.27_{\pm4.74}$ | $\mathbf{107.28}_{\pm3.92}$ | 111.33 |
| | halfcheetah | $-0.93_{\pm0.35}$ | $2.24_{\pm0.01}$ | $76.92_{\pm7.53}$ | $3.19_{\pm1.82}$ | $4.02_{\pm1.74}$ | $43.65_{\pm3.85}$ | $\mathbf{88.77}_{\pm1.26}$ | 88.83 |
| | walker2d | $0.98_{\pm0.83}$ | $0.74_{\pm0.20}$ | $83.23_{\pm19.00}$ | $3.93_{\pm0.76}$ | $36.11_{\pm9.19}$ | $\mathbf{108.35}_{\pm0.13}$ | $\mathbf{108.40}_{\pm0.21}$ | 106.92 |
| | ant | $0.91_{\pm3.93}$ | $35.38_{\pm2.66}$ | $67.14_{\pm8.30}$ | $6.59_{\pm6.86}$ | $99.90_{\pm2.59}$ | $110.79_{\pm1.33}$ | $\mathbf{120.09}_{\pm5.17}$ | 130.75 |
| medium+ expert | hopper | $16.09_{\pm12.80}$ | $59.25_{\pm3.71}$ | $88.51_{\pm16.73}$ | $55.74_{\pm2.10}$ | $65.99_{\pm8.05}$ | $108.22_{\pm1.95}$ | $\mathbf{110.20}_{\pm0.51}$ | 111.33 |
| | halfcheetah | $-1.79_{\pm0.22}$ | $42.45_{\pm0.42}$ | $81.15_{\pm2.84}$ | $53.80_{\pm4.18}$ | $58.74_{\pm1.84}$ | $88.54_{\pm3.77}$ | $\mathbf{93.12}_{\pm0.32}$ | 88.83 |
| | walker2d | $2.43_{\pm1.82}$ | $72.76_{\pm3.82}$ | $108.54_{\pm1.81}$ | $6.91_{\pm0.71}$ | $105.41_{\pm0.33}$ | $86.59_{\pm12.32}$ | $\mathbf{108.12}_{\pm0.13}$ | 106.92 |
| | ant | $0.86_{\pm7.42}$ | $95.47_{\pm10.37}$ | $120.36_{\pm7.67}$ | $104.00_{\pm3.62}$ | $108.14_{\pm1.90}$ | $98.46_{\pm1.44}$ | $\mathbf{124.72}_{\pm3.49}$ | 130.75 |
| medium few-expert | hopper | $7.37_{\pm1.13}$ | $46.87_{\pm5.31}$ | $50.01_{\pm10.36}$ | $53.50_{\pm1.55}$ | $57.24_{\pm3.03}$ | $96.95_{\pm7.89}$ | $\mathbf{108.96}_{\pm1.33}$ | 111.33 |
| | halfcheetah | $-1.15_{\pm0.06}$ | $42.21_{\pm0.06}$ | $75.96_{\pm4.54}$ | $42.88_{\pm0.63}$ | $27.85_{\pm6.03}$ | $59.40_{\pm6.80}$ | $\mathbf{89.47}_{\pm0.82}$ | 88.83 |
| | walker2d | $2.02_{\pm0.72}$ | $70.42_{\pm2.86}$ | $91.25_{\pm17.63}$ | $9.08_{\pm3.67}$ | $75.22_{\pm7.05}$ | $74.35_{\pm0.80}$ | $\mathbf{108.15}_{\pm0.43}$ | 106.92 |
| | ant | $-10.45_{\pm1.63}$ | $81.63_{\pm6.67}$ | $110.38_{\pm10.96}$ | $88.20_{\pm1.13}$ | $90.34_{\pm2.56}$ | $90.77_{\pm0.50}$ | $\mathbf{120.36}_{\pm1.25}$ | 130.75 |
| cloned+expert | pen | $13.95_{\pm11.04}$ | $34.94_{\pm11.10}$ | $95.04_{\pm4.48}$ | $15.71_{\pm11.36}$ | $23.39_{\pm4.56}$ | $26.48_{\pm3.33}$ | $\mathbf{82.77}_{\pm4.84}$ | 106.42 |
| | door | $-0.22_{\pm0.05}$ | $0.011_{\pm0.00}$ | $102.75_{\pm4.05}$ | $1.57_{\pm2.32}$ | $0.07_{\pm0.14}$ | $93.29_{\pm13.65}$ | $\mathbf{102.77}_{\pm0.96}$ | 103.94 |
| | hammer | $2.41_{\pm4.48}$ | $5.45_{\pm7.84}$ | $95.77_{\pm17.90}$ | $1.07_{\pm1.30}$ | $1.29_{\pm0.12}$ | $\mathbf{91.80}_{\pm22.17}$ | $\mathbf{94.59}_{\pm9.39}$ | 125.71 |
| human+expert | pen | $13.83_{\pm10.76}$ | $90.76_{\pm25.09}$ | $103.72_{\pm2.90}$ | $58.62_{\pm7.52}$ | $-2.56_{\pm1.30}$ | $31.95_{\pm7.43}$ | $\mathbf{95.77}_{\pm8.91}$ | 106.42 |
| | door | $-0.03_{\pm0.05}$ | $103.71_{\pm1.22}$ | $104.70_{\pm0.55}$ | $29.84_{\pm12.17}$ | $0.15_{\pm0.02}$ | $0.11_{\pm0.40}$ | $\mathbf{100.77}_{\pm1.68}$ | 103.94 |
| | hammer | $0.18_{\pm0.14}$ | $122.61_{\pm4.85}$ | $125.19_{\pm3.29}$ | $33.28_{\pm16.83}$ | $2.02_{\pm0.77}$ | $6.93_{\pm2.45}$ | $\mathbf{93.34}_{\pm7.41}$ | 125.71 |
| partial+expert | kitchen | $2.5_{\pm5.0}$ | $45.5_{\pm1.87}$ | $60.0_{\pm5.70}$ | $36.67_{\pm5.77}$ | $12.33_{\pm5.38}$ | $23.00_{\pm25.87}$ | $\mathbf{58.95}_{\pm2.27}$ | 75.0 |
| mixed+expert | kitchen | $2.2_{\pm3.8}$ | $42.1_{\pm1.12}$ | $52.0_{\pm1.0}$ | $\mathbf{48.33}_{\pm6.29}$ | $7.50_{\pm4.16}$ | $29.17_{\pm13.97}$ | $46.45_{\pm0.84}$ | 75.0 |

Table 2: Average normalized return over last 10 evaluations of IOSTOM against baselines on the D4RL suboptimal datasets with 1 expert trajectory. The mean and std are obtained over 5 random seeds. LfO methods with avg. perf within the std-dev of the top performing LfO approach is in **bold**.

.

ronments: 4 locomotion (Hopper, HalfCheetah, Walker2d, Ant) and 4 manipulation (Pen, Door, Hammer, Kitchen) [42]. Each task's expert dataset contains one trajectory. Suboptimal datasets for locomotion mix D4RL 'random' or 'medium' data with 200 ('expert') or 30 ('few-expert') expert trajectories. For manipulation, D4RL non-expert datasets ('mixed' and 'partial' for Kitchen; 'human' and 'cloned' for others) are mixed with up to 30 expert trajectories. This results in 24 diverse tasks for comparing IOSTOM against baselines, with manipulation tasks being more challenging due to larger state spaces. More details on environment and dataset are included in the Appendix.

Table 2 presents results for IOSTOM and baselines. We also include the results of some Learning from Demonstration (LfD) methods such as Behavior Cloning (BC) and ReCOIL [39] to serve as the reference upper bound of LfO methods because they have access to expert actions during learning. We choose these two approaches as ReCOIL is the SOTA offline LfD method while BC is the most popular IL algorithm. Their results are taken directly from ReCOIL's paper which uses a similar setting. As shown in Table 2, IOSTOM leads on 23/24 tasks, only marginally underperforming DILO on 'walker2d random+expert' while still matching expert performance. Discriminator-based methods (SMODICE, PW-DICE) degrade significantly with few expert examples or on high-dimensional manipulation tasks due to discriminator overfitting. While DILO's discriminator-free nature mitigates this, it still struggles in 'few-expert' settings (e.g., 'halfcheetah') and 'human+expert' tasks where training can diverge (see Appendix for further discussion). BC methods with access to expert actions also exhibit poor performance on most tasks. Notably, IOSTOM's performance is comparable to, and sometimes surpasses, ReCOIL on locomotion tasks, showcasing its effectiveness and potential to bridge the gap between LfD and LfO.

## 6.2 LfO with subsampled expert

This section focuses on benchmarking the sample efficiency of our approach (Question (**Q3**)). We adapt the subsampled expert trajectory setting from LfD literature [13, 20]) to construct a subsampled state-only expert dataset. Specifically, expert trajectories are sub-sampled by keeping a transition every 20 time steps (i.e. subsampling rate is 20) starting with a random offset. This process will create incomplete expert trajectories which makes both BC and DICE method like ValueDICE [21] fail as shown in [51]. This setting may not be valid in case of DILO because it requires the triplet $(s, s', s'')$ which is equivalent to two transitions inside action-labeled expert dataset; we still adapt the 2-transition version of the subsampling procedure only for DILO. LobsDICE also considers the similar setting for LfO like us on locomotion tasks, but they construct $D^E$ using 50 sub-sampled expert trajectories, which means using $\frac{50}{20} = 2.5$ times of total transitions of an expert trajectory. This makes this setting still easy to deal with for both our approach and baselines. Therefore, we

| Suboptimal Dataset | Env | SMODICE (full) | SMODICE (sub) | PW-DICE (full) | PW-DICE (sub) | DILO (full) | DILO (sub) | IOSTOM (full) | IOSTOM (sub) |
|---|---|---|---|---|---|---|---|---|---|
| random+ expert | hopper | $106.56_{\pm0.53}$ | $108.33_{\pm0.43}$ | $108.09_{\pm2.39}$ | $97.35_{\pm2.72}$ | $86.35_{\pm38.00}$ | $13.25_{\pm12.95}$ | $109.32_{\pm1.08}$ | **$109.94_{\pm0.46}$** |
| | halfcheetah | $85.55_{\pm1.39}$ | $78.63_{\pm5.04}$ | $86.11_{\pm4.39}$ | $37.95_{\pm9.94}$ | $91.53_{\pm0.27}$ | $92.06_{\pm0.29}$ | $93.02_{\pm0.40}$ | **$93.23_{\pm0.24}$** |
| | walker2d | $107.93_{\pm1.26}$ | $107.46_{\pm0.51}$ | $107.48_{\pm0.53}$ | $101.59_{\pm1.11}$ | $108.31_{\pm0.18}$ | $41.98_{\pm35.80}$ | $107.98_{\pm0.20}$ | **$108.01_{\pm0.16}$** |
| | ant | $126.08_{\pm0.73}$ | $124.13_{\pm3.74}$ | $126.89_{\pm1.17}$ | $112.99_{\pm6.28}$ | $125.39_{\pm2.37}$ | $30.27_{\pm2.47}$ | $128.19_{\pm1.52}$ | **$126.23_{\pm2.87}$** |
| random+ few-expert | hopper | $58.30_{\pm9.96}$ | $58.44_{\pm10.26}$ | $75.04_{\pm14.21}$ | $48.30_{\pm20.09}$ | $104.27_{\pm4.74}$ | $92.52_{\pm10.81}$ | $107.28_{\pm3.92}$ | **$105.20_{\pm5.90}$** |
| | halfcheetah | $3.19_{\pm1.82}$ | $3.06_{\pm1.29}$ | $4.02_{\pm1.74}$ | $3.91_{\pm1.17}$ | $43.65_{\pm3.85}$ | $44.22_{\pm4.09}$ | $88.77_{\pm1.26}$ | **$86.09_{\pm3.82}$** |
| | walker2d | $3.93_{\pm0.76}$ | $4.78_{\pm2.32}$ | $36.11_{\pm9.19}$ | $26.29_{\pm10.97}$ | $108.35_{\pm0.13}$ | $33.69_{\pm5.97}$ | $108.40_{\pm0.21}$ | **$104.32_{\pm8.62}$** |
| | ant | $6.59_{\pm6.86}$ | $6.33_{\pm3.12}$ | $99.90_{\pm2.59}$ | $82.81_{\pm8.88}$ | $110.79_{\pm1.33}$ | $31.91_{\pm0.65}$ | $120.09_{\pm5.17}$ | **$123.83_{\pm4.29}$** |
| medium+ expert | hopper | $55.74_{\pm2.10}$ | $54.24_{\pm2.47}$ | $65.99_{\pm8.05}$ | $63.03_{\pm7.24}$ | $108.22_{\pm1.95}$ | $54.42_{\pm0.47}$ | $110.20_{\pm0.51}$ | **$109.72_{\pm0.92}$** |
| | halfcheetah | $53.80_{\pm4.18}$ | $50.06_{\pm1.87}$ | $58.74_{\pm1.84}$ | $62.78_{\pm2.70}$ | $88.54_{\pm3.77}$ | $42.61_{\pm0.21}$ | $93.12_{\pm0.32}$ | **$92.97_{\pm0.35}$** |
| | walker2d | $6.91_{\pm0.71}$ | $1.77_{\pm1.63}$ | $105.41_{\pm0.33}$ | $82.90_{\pm18.45}$ | $86.59_{\pm12.32}$ | $83.41_{\pm24.57}$ | $108.12_{\pm0.13}$ | **$108.59_{\pm0.17}$** |
| | ant | $104.00_{\pm3.62}$ | $99.52_{\pm1.18}$ | $108.14_{\pm1.90}$ | $110.68_{\pm4.20}$ | $98.46_{\pm1.44}$ | $105.73_{\pm5.35}$ | $124.72_{\pm3.49}$ | **$124.06_{\pm1.66}$** |
| medium few-expert | hopper | $53.50_{\pm1.55}$ | $54.26_{\pm1.09}$ | $57.24_{\pm3.03}$ | $50.51_{\pm4.21}$ | $96.95_{\pm7.89}$ | $55.50_{\pm1.33}$ | $108.96_{\pm1.33}$ | **$107.21_{\pm1.69}$** |
| | halfcheetah | $42.88_{\pm0.63}$ | $42.88_{\pm0.74}$ | $27.85_{\pm6.03}$ | $11.99_{\pm5.61}$ | $59.40_{\pm6.80}$ | $53.53_{\pm7.52}$ | $89.47_{\pm0.82}$ | **$87.45_{\pm3.67}$** |
| | walker2d | $9.08_{\pm3.67}$ | $3.05_{\pm2.22}$ | $75.22_{\pm7.05}$ | $52.95_{\pm11.10}$ | $74.35_{\pm0.80}$ | $54.55_{\pm2.89}$ | $108.15_{\pm0.43}$ | **$108.45_{\pm0.30}$** |
| | ant | $88.20_{\pm1.13}$ | $88.80_{\pm5.18}$ | $90.34_{\pm2.56}$ | $89.69_{\pm2.23}$ | $90.77_{\pm0.50}$ | $90.90_{\pm1.49}$ | $120.36_{\pm1.25}$ | **$117.29_{\pm1.85}$** |

Table 3: Comparison of normalized returns obtained by different offline LfO methods on expert dataset with 1 expert trajectory denoted as (full) or 5 subsampled expert trajectories (subsampling rate is 20) denoted as (sub). The mean and std are obtained over 5 random seeds. Methods on subsampled expert dataset with avg. perf within the std-dev of the top performing method is in **bold**. Methods with greater than 5% performance decrease on subsampled expert datasets are highlighted in blue.

construct $D^E$ from 5 subsampled trajectories only (i.e. 0.25x total transitions of an expert trajectory) and evaluate all LfO methods on locomotion tasks with the same suboptimal dataset in Section 6.1.

Table 3 shows the comparison results on the subsampled setting. IOSTOM continues to outperform all baselines on these challenging tasks. Furthermore, its performance does not change much compared to using complete expert trajectory even when the total number of expert samples is reduced by 4 times. SMODICE is also robust on 12/16 tasks but its performance on 'few-expert' setting is still poor. Both DILO and PW-DICE face a large drop (>5 %) on the performance of most tasks in the scenario of less samples and incomplete trajectories.

## 6.3  LfO with mismatched expert

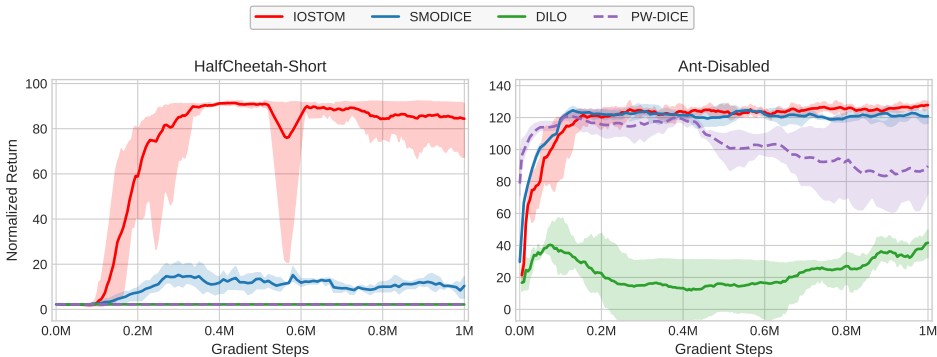

Figure 1: Comparison results for LfO with mismatched experts

To evaluate IOSTOM performance when learning from experts of different dynamics, we adopt SMODICE's mismatched dynamics setting. We test on 'HalfCheetah-Short' (halved torso) and 'Ant-Disabled' (partially amputated front leg) (See Appendix H of SMODIE [27] for illustration), using one expert trajectory from these modified agents. The suboptimal dataset remains the 'random+expert' data (Section 6.1) from the original agents. This setting creates a clear mismatch between expert and interaction datasets. Figure 1 shows IOSTOM outperforming baselines on these challenging tasks, while DILO performs worst. The poor performance of DILO can be due to the use of visitation distribution $d(s, s', a')$ in its objective which matches the wrong $a'$ in the mismatched expert dataset.

## 6.4  LfO for marine navigation

We next test IOSTOM in a real-world domain, the maritime navigation problem. Our goal is to learn IL policies that can behave like human experts (ship pilots) for navigating vessels (mainly large tankers and cargos). These polices offer significant benefits for operational safety and efficiency.

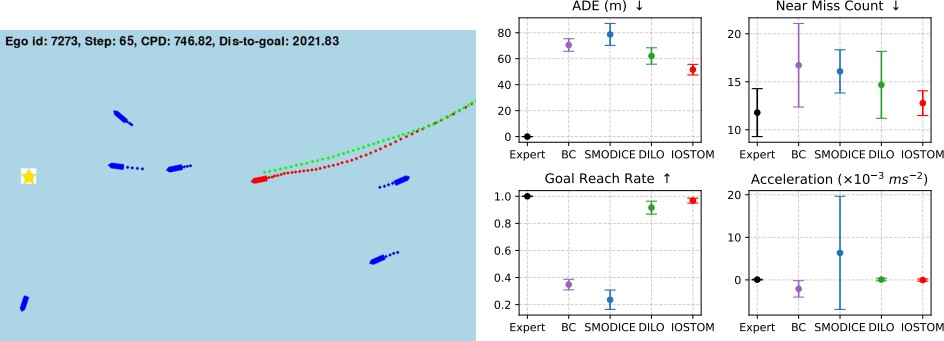

Figure 2: **Left**: Visualization of an episode from our maritime simulator. Blue vessels follows their historical trajectory; red vessel is controlled by IL policy (destination ⋆ marker), green dots denote historical trajectory of ego vessel. **Right**: Comparison results on various performance metrics. Details on metrics are in the Appendix.

For instance, they can be integrated into Vessel Traffic Information Systems to provide port watch operators with accurate short-term predictions of vessel movements, especially in congested ports such as Singapore strait. Learned IL policies also enable to do what-if analysis such as how would safety be affected when there is traffic surge (e.g., by simulating additional vessel arrivals and assigning them IL policies). Prior RL-based approaches to maritime traffic management [40, 41] rely on online learning, requiring costly simulator interactions and accurate simulation of vessel dynamics. In contrast, our IL method directly learns vessel behavior from large offline datasets, offering easier and more accurate modeling as shown in our results (see later metrics such as ADE, goal rate).

We collect a large amount of historical navigation data ($\sim$ 2 years) of vessels operating in a hotspot region in Singpaore strait (among top 5 busiest ports) recorded in the Automatic Identification System (AIS). We use these data with **ShipNaviSim**, a data-driven maritime traffic simulator [32] to construct a realistic environment featuring an IOSTOM-controlled ego agent (red) and log-play agents (blue) that follow their actual trajectories, as illustrated in Figure 2. The ego agent controlled by IL policy tries to reach the goal ('⋆') while avoiding collisions with other log-play agents. The ego agent can also observe past states (blue and red dots) of its and close surrounding agents (its observation space). Because AIS data does not contain any action information, we use an inverse kinematics model (IVM) to construct the action space and generate action for AIS data. The state-only expert dataset in this setting is easy to obtain due to the action-free nature of AIS data. We generate the suboptimal dataset by adding random noise action-labeled expert trajectories. Further details about environment and dataset generation can be found in the Appendix.

We evaluate our approach in maritime navigation setting against BC, SMODICE, and DILO. Results are shown in Figure 2 using metrics relevant to this domain, introduced in [32], which reflect how well the learned agent imitates expert behavior. **ADE** (Average Displacement Error; lower is better) measures how far, on average, the agent's trajectory deviates from the expert's. **Goal reach rate** (higher is better) indicates how often the ship reaches the goal. **Near-miss count** captures the number of close-quarter situations, defined as scenarios where two ships come close to each other posing a collision risk; lower values indicate reduced collision risk, and **average acceleration** should closely match that of the expert. Mean and standard deviations are over 5 seeds for each method. Our approach outperforms across all three baselines in ADE, near-miss count, and goal reach rate, while maintaining an acceleration profile similar to the expert. DILO is the second-best performer. SMODICE struggles due to high-dimensional observation space—which includes nearby ships and trajectory history; leading to a poorly trained state discriminator and worse performance than BC.

# 7 Conclusion

We presented IOSTOM, a discriminator-free Q-learning framework for offline imitation learning from observations. By learning an implicit policy in the form of state-to-state transitions and matching joint state visitation distributions, our method avoids reliance on action labels for value function learning (Q-learning) and eliminates the need for inverse dynamics models in policy extraction. Extensive experimental results and ablation studies demonstrate that IOSTOM achieves strong empirical performance and improves sample efficiency compared to prior approaches.

## Acknowledgments and Disclosure of Funding

This research/project is supported by the National Research Foundation, Singapore and DSO National Laboratories under the AI Singapore Programme (AISG Award No: AISG2-RP-2020-017).

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

# APPENDIX

## Contents

## A    Limitations and Future Work

Despite its strong empirical performance, IOSTOM has some limitations. First, like most LfO methods, it assumes access to high-quality consecutive state pairs $(s, s')$, which may not always be available in real-world datasets. Second, we assume that actions are fully observable in the sub-optimal dataset, which might not hold in practice. While these limitations are beyond the scope of this work, they highlight important directions for future research.

## B    Missing Proofs and Derivations

### B.1    Proof of Proposition 1

**Proposition.** *The joint visitation distribution $d^g(s, s')$ can be expressed as:*

$$d^g(s, s') = (1 - \gamma)d_0(s)g(s' \mid s) + \gamma g(s' \mid s) \sum_{\bar{s}} d^g(\bar{s}, s) \tag{10}$$

*Proof.* We recall that $d^\pi(s, a) = (1 - \gamma)d_0(s)\pi(a \mid s) + \gamma \sum_{s', a'} d^\pi(s', a')\mathcal{T}(s \mid s', a')\pi(a \mid s)$. Therefore, we have:

$$
\begin{aligned}
d^g(s, s') &= \sum_a d^\pi(s, a)\mathcal{T}(s' \mid s, a) \\
&= \sum_a (1 - \gamma)d_0(s)\pi(a \mid s)\mathcal{T}(s' \mid s, a) + \sum_a \gamma \mathcal{T}(s' \mid s, a) \sum_{\bar{s}, \bar{a}} d^\pi(\bar{s}, \bar{a})\mathcal{T}(s \mid \bar{s}, \bar{a})\pi(a \mid s) \\
&= (1 - \gamma)d_0(s)g(s' \mid s) + \gamma g(s' \mid s) \sum_{\bar{s}, \bar{a}} d^\pi(\bar{s}, \bar{a})\mathcal{T}(s \mid \bar{s}, \bar{a}) \\
&= (1 - \gamma)d_0(s)g(s' \mid s) + \gamma g(s' \mid s) \sum_{\bar{s}} d^g(\bar{s}, s).
\end{aligned}
$$

as desired.    □

### B.2    Proof of Proposition 2

**Proposition.** *The constrained optimization problem in Equation* (2) *is equivalent to the following unconstrained max-min problem:*

$$
\begin{aligned}
\max_g \min_{Q(s, s')} \Big\{ &\alpha(1 - \gamma) \mathbb{E}_{(s, s') \sim d_0} \left[ Q(s, s') \right] + \mathbb{E}_{(s, s') \sim d_{mix}^{E, I}} \left[ f^* \left( \gamma \mathbb{E}_{s'' \sim g(\cdot|s')} \left[ Q(s', s'') \right] - Q(s, s') \right) \right] \\
&- (1 - \alpha) \mathbb{E}_{(s, s') \sim d^I} \left[ \gamma \mathbb{E}_{s'' \sim g(\cdot|s')} \left[ Q(s', s'') \right] - Q(s, s') \right] \Big\},
\end{aligned}
$$

*where $Q(s, s')$ are the Lagrange multipliers and $f^*$ denotes the convex conjugate of a chosen convex function $f$.*

*Proof.* We recall that the primal formulation in Equation (2) is as follows:

$$
\begin{aligned}
\max_{g, d \geq 0} \quad & -\mathbb{D}_f(d_{mix}^I(s, s') \| d_{mix}^{E, I}(s, s')) \\
s.t. \quad & d(s, s') = (1 - \gamma)d_0(s)g(s'|s) + \gamma g(s'|s) \sum_{\bar{s}} d(\bar{s}, s)
\end{aligned}
$$

We first apply duality on the inner maximization problem of the above formulation:

$$\max_{g,d\geq 0} \min_{Q(s,s')} -\mathbb{D}_f(d_{mix}^I(s,s')\|d_{mix}^{E,I}(s,s'))$$

$$+ \alpha \sum_{s,s'} Q(s,s')\left((1-\gamma)d_0(s).g(s'|s) + \gamma g(s'|s)\sum_{\bar{s}} d(\bar{s},s) - d(s,s')\right) \tag{11}$$

$$= \max_{\pi,d\geq 0} \min_{Q(s,s')} \alpha(1-\gamma)\mathbb{E}_{d_0(s),g(s'|s)}\left[Q(s,s')\right]$$

$$+ \alpha\mathbb{E}_{s,s'\sim d}\left[\gamma\sum_{s''} g(s''|s')Q(s',s'') - Q(s,s')\right] - \mathbb{D}_f(d_{mix}^I(s,s')\|d_{mix}^{E,I}(s,s')) \tag{12}$$

Step (11) to (12) is equivalent to changing the following order of summation:

$$\sum_{s,s'} Q(s,s')g(s'|s)\sum_{\bar{s}} d(\bar{s},s)$$

$$= \sum_{\bar{s},s} d(\bar{s},s)\sum_{s'} Q(s,s')g(s'|s)$$

$$= \sum_{s,s'} d(s,s')\sum_{s''} Q(s',s'')g(s''|s')$$

By adding and subtracting another term below, (12) becomes:

$$= \max_{g,d\geq 0} \min_{Q(s,s')} \alpha(1-\gamma)\mathbb{E}_{d_0(s),g(s'|s)}\left[Q(s,s')\right]$$

$$+ \alpha\mathbb{E}_{s,s'\sim d}\left[\gamma\sum_{s''} g(s''|s')Q(s',s'') - Q(s,s')\right]$$

$$+ (1-\alpha)\mathbb{E}_{s,s'\sim d^I}\left[\gamma\sum_{s''} g(s''|s')Q(s',s'') - Q(s,s')\right]$$

$$- (1-\alpha)\mathbb{E}_{s,s'\sim d^I}\left[\gamma\sum_{s''} g(s''|s')Q(s',s'') - Q(s,s')\right]$$

$$- \mathbb{D}_f(d_{mix}^I(s,s')\|d_{mix}^{E,I}(s,s')) \tag{13}$$

We can swap $\max_d$ and $\min_Q$ in (13) due to strong duality.

$$(13) = \max_g \min_{Q(s,s')} \max_{d_{mix}^I(s,s')\geq 0} \alpha(1-\gamma)\mathbb{E}_{d_0(s),g(s'|s)}\left[Q(s,s')\right]$$

$$+ \mathbb{E}_{s,s'\sim d_{mix}^I}\left[\gamma\sum_{s''} g(s''|s')Q(s',s'') - Q(s,s')\right] - \mathbb{D}_f(d_{mix}^I(s,s')\|d_{mix}^{E,I}(s,s'))$$

$$- (1-\alpha)\mathbb{E}_{s,s'\sim d^I}\left[\gamma\sum_{s''} g(s''|s')Q(s',s'') - Q(s,s')\right] \tag{14}$$

$$= \max_g \min_{Q(s,s')} \alpha(1-\gamma)\mathbb{E}_{d_0(s),g(s'|s)}\left[Q(s,s')\right]$$

$$+ \mathbb{E}_{s,s'\sim d_{mix}^{E,I}}\left[f^*\left(\gamma\sum_{s''} g(s''|s')Q(s',s'') - Q(s,s')\right)\right]$$

$$- (1-\alpha)\mathbb{E}_{s,s'\sim d^I}\left[\gamma\sum_{s''} g(s''|s')Q(s',s'') - Q(s,s')\right] \tag{15}$$

where $f^*$ is convex conjugate of convex $f$-divergence function. (14) to (15) can be proved by the following equation using the interchangeability principle [5]:

$$\max_{d^I_{mix}(s,s') \geq 0} \mathbb{E}_{s,s' \sim d^I_{mix}} \left[ \gamma \sum_{s''} g(s''|s') Q(s', s'') - Q(s, s') \right] - \mathbb{D}_f(d^I_{mix}(s, s') \| d^{E,I}_{mix}(s, s'))$$

$$= \max_{d^I_{mix}(s,s') \geq 0} \mathbb{E}_{s,s' \sim d^{E,I}_{mix}} \left[ \frac{d^I_{mix}(s, s')}{d^{E,I}_{mix}(s, s')} \left( \gamma \sum_{s''} g(s''|s') Q(s', s'') - Q(s, s') \right) - f\left( \frac{d^I_{mix}(s, s')}{d^{E,I}_{mix}(s, s')} \right) \right]$$

$$= \mathbb{E}_{s,s' \sim d^{E,I}_{mix}} \left[ f^* \left( \gamma \sum_{s''} g(s''|s') Q(s', s'') - Q(s, s') \right) \right]$$

Finally, the objective (15) is the unconstrained dual problem of Equation (2). $\qquad \square$

## B.3 Complete derivation of transforming objective function (3) to (4)

$$\alpha(1-\gamma)\mathbb{E}_{(s,s') \sim d_0} [Q(s, s')] + \mathbb{E}_{(s,s') \sim d^{E,I}_{mix}} \left[ f^* \left( \gamma \mathbb{E}_{s'' \sim g(\cdot|s')} [Q(s', s'')] - Q(s, s') \right) \right]$$
$$- (1-\alpha) \mathbb{E}_{(s,s') \sim d^I} \left[ \gamma \mathbb{E}_{s'' \sim g(\cdot|s')} [Q(s', s'')] - Q(s, s') \right] \tag{16}$$

$$= \alpha(1-\gamma)\mathbb{E}_{(s,s') \sim d_0} [Q(s, s')] + \mathbb{E}_{(s,s') \sim d^{E,I}_{mix}} \left[ f^* \left( \gamma Q(s', g) - Q(s, s') \right) \right]$$
$$- (1-\alpha) \mathbb{E}_{(s,s') \sim d^I} \left[ \gamma Q(s', g) - Q(s, s') \right] \tag{17}$$

$$= \alpha(1-\gamma)\mathbb{E}_{(s,s') \sim d_0} [Q(s, s')] + \alpha\mathbb{E}_{(s,s') \sim d^E} \left[ f^* \left( \gamma Q(s', g) - Q(s, s') \right) \right]$$
$$+ (1-\alpha)\mathbb{E}_{(s,s') \sim d^I} \left[ f^* \left( \gamma Q(s', g) - Q(s, s') \right) \right] - (1-\alpha) \mathbb{E}_{(s,s') \sim d^I} \left[ \gamma Q(s', g) - Q(s, s') \right] \tag{18}$$

$$= \alpha(1-\gamma)\mathbb{E}_{(s,s') \sim d_0} [Q(s, s')] + \alpha\mathbb{E}_{(s,s') \sim d^E} \left[ f^* \left( \gamma Q(s', g) - Q(s, s') \right) \right]$$
$$+ (1-\alpha)\mathbb{E}_{(s,s') \sim d^I} \left[ f^* \left( \gamma Q(s', g) - Q(s, s') \right) - \left( \gamma Q(s', g) - Q(s, s') \right) \right] \tag{19}$$

$$= \alpha(1-\gamma)\mathbb{E}_{(s,s') \sim d_0} [Q(s, s')] + \alpha\mathbb{E}_{(s,s') \sim d^E} \left[ f^* \left( \gamma Q(s', g) - Q(s, s') \right) \right]$$
$$+ (1-\alpha) \mathbb{E}_{(s,s') \sim d^I} \left[ \widetilde{f^*} \left( \gamma Q(s', g) - Q(s, s') \right) \right] \tag{20}$$

(16) to (17) by defining $Q(s, g) = \mathbb{E}_{s' \sim g(\cdot|s)} [Q(s, s')]$. (17) to (18) due to $d^{E,I}_{mix}(s, s') = \alpha d^E(s, s') + (1-\alpha)d^I(s, s')$. (19) to (20) by defining $\widetilde{f^*}(t) = f^*(t) - t$.

## B.4 Proof of Proposition 3

**Proposition.** $\max_g \{V^g_Q(s)\}$ *can be approximated via the following Extreme-V objective:*

$$\min_V \left\{ J(V \mid Q) = \mathbb{E}_{(s,s') \sim d^{E,I}_{mix}} \left[ \exp(\omega(s, s')) + \omega(s, s') - 1 \right] \right\}.$$

*where* $\omega(s, s') = (Q(s, s') - V(s))/\beta$.

*Proof.* Recall that:

$$V^g_Q(s) = \mathbb{E}_{s' \sim g(\cdot|s)} \left[ Q(s, s') - \beta \log \frac{g(s' \mid s)}{\mu(s' \mid s)} \right],$$

which is the expected reward under transition distribution $g(\cdot \mid s)$, regularized by the KL divergence from a reference distribution $\mu(\cdot \mid s)$. Moreover, the problem $\max_g \left\{ V^g_Q(s) \right\}$ is a classic entropy-regularized expected reward maximization problem. The optimal solution has a closed form:

$$\max_g \left\{ V^g_Q(s) \right\} = \beta \log \sum_{s'} \mu(s' \mid s) \exp\left( \frac{Q(s, s')}{\beta} \right). \tag{21}$$

We now write the function $J(V \mid Q)$ as:

$$J(V \mid Q) = \sum_{(s,s')} \mu(s' \mid s) \left[ \exp\left( \frac{Q(s, s') - V(s)}{\beta} \right) + \frac{Q(s, s') - V(s)}{\beta} - 1 \right].$$

For any state $s$, and fixed $Q$, the function $J(V \mid Q)$ is convex in $V(s)$ because:

- The exponential function $\exp\left(\frac{Q(s,s')-V(s)}{\beta}\right)$ is convex in $V(s)$,

- The linear term $(Q(s,s')-V(s))/\beta$ is also convex (affine),

- The sum and non-negative weights preserve convexity.

To find the minimum of $J(V \mid Q)$ with respect to $V$, we take the derivative with respect to $V(s)$ and set it to zero:

$$\frac{\partial J(V \mid Q)}{\partial V(s)} = \sum_{s'} \mu(s' \mid s) \left[ -\frac{1}{\beta} \exp\left(\frac{Q(s,s')-V(s)}{\beta}\right) - \frac{1}{\beta} \right] = 0.$$

Rewriting:

$$\sum_{s'} \mu(s' \mid s) \exp\left(\frac{Q(s,s')-V(s)}{\beta}\right) = \sum_{s'} \mu(s' \mid s).$$

We have $\sum_{s'} \mu(s' \mid s) = 1$, this gives:

$$\sum_{s'} \mu(s' \mid s) \exp\left(\frac{Q(s,s')-V(s)}{\beta}\right) = 1.$$

Bringing the constant outside the exponential:

$$\exp\left(-\frac{V(s)}{\beta}\right) \sum_{s'} \mu(s' \mid s) \exp\left(\frac{Q(s,s')}{\beta}\right) = 1,$$

$$\Rightarrow \exp\left(-\frac{V(s)}{\beta}\right) = \frac{1}{\sum_{s'} \mu(s' \mid s) \exp\left(\frac{Q(s,s')}{\beta}\right)},$$

Taking the logarithm of both sides and solving for $V(s)$, we obtain the closed-form solution to $\min_V J(V|Q)$ as:

$$V^*(s) = \beta \log \sum_{s'} \mu(s' \mid s) \exp\left(\frac{Q(s,s')}{\beta}\right). \tag{22}$$

Combined (21) and (22) we get:

$$V^*(s) = \max_g \{V_Q^g(s)\}$$

as desired. $\qquad\square$

### B.5  Proof of Proposition 4

**Proposition.** *Under any convex function $f$, $L(Q,V)$ is concave in $Q$, and the Extreme-V loss $J(V \mid Q)$ is convex in $V$.*

*Proof.* We rewrite the objective $L(Q,V)$ as:

$$\min_Q L(Q,V) = \alpha(1-\gamma)\,\mathbb{E}_{(s,s')\sim\mathcal{D}_0}\left[Q(s,s')\right] + \alpha\mathbb{E}_{(s,s')\sim\mathcal{D}^E}\left[f^*\left(\gamma V(s') - Q(s,s')\right)\right]$$

$$+ (1-\alpha)\mathbb{E}_{(s,s')\sim\mathcal{D}^I}\left[\widetilde{f^*}\left(\gamma V(s') - Q(s,s')\right)\right].$$

We now analyze the convexity of $L(Q,V)$ with respect to $Q$. Note the following:

- The first term, $\mathbb{E}_{\mathcal{D}_0}[Q(s,s')]$, is linear in $Q$, and hence convex.

- The functions $f^*$ and $\widetilde{f^*}$ are convex (as they are convex conjugates of proper convex functions).

- The composition of a convex function with an affine function (i.e., $\gamma V(s') - Q(s,s')$) is convex in $Q$.

- Expectations of convex functions preserve convexity.

Therefore, each term in $L(Q, V)$ is convex in $Q$, and the entire objective $L(Q, V)$ is convex in $Q$, as desired.

The convexity of $J(V \mid Q)$ in $V$ follows directly from the discussion in the proof of Proposition (3).

$\square$

## B.6  Proof of Proposition 5

**Proposition.** *The following Q-weighted behavior cloning objective returns the same optimal implicit policy as the original advantage-weighted BC formulation:*

$$\max_g \ \mathbb{E}_{(s,s') \sim d^I} \left[ \exp\left( \tau Q(s, s') \right) \log g(s' \mid s) \right]$$

*Proof.* We write the objective function as:

$$F(g) = \sum_{(s,s')} \mu^I(s' \mid s) \exp\left( \tau Q(s, s') \right) \log g(s' \mid s),$$

where $\mu^I(s' \mid s)$ is the state-to-state transition probability (i.e., the implicit behavior policy) for the dataset $\mathcal{D}^I$. For each fixed state $s$, the expression

$$\sum_{s'} \mu^I(s' \mid s) \exp\left( \tau Q(s, s') \right) \log g(s' \mid s)$$

is a weighted log-likelihood, where the weights $\mu^I(s' \mid s) \exp(\tau Q(s, s'))$ are known. Maximizing this with respect to $g(\cdot \mid s)$ under the constraint that $g(\cdot \mid s)$ is a valid probability distribution (i.e., $\sum_{s'} g(s' \mid s) = 1$) leads to a standard result from maximum likelihood estimation with importance weights. The closed-form solution is:

$$g^*(s' \mid s) = \frac{\mu^I(s' \mid s) \exp\left( \tau Q(s, s') \right)}{\sum_{s''} \mu^I(s'' \mid s) \exp\left( \tau Q(s, s'') \right)}.$$

We now consider the advantage-weighted behavior cloning objective:

$$\max_g \ \mathbb{E}_{(s,s') \sim d^I} \left[ \exp\left( \tau(Q(s, s') - V(s)) \right) \log g(s' \mid s) \right],$$

In a similar fashion to soft behavior cloning, this yields the following closed-form optimal "implicit policy":

$$g^{**}(s' \mid s) = \frac{\mu^I(s' \mid s) \exp\left( \tau(Q(s, s') - V(s)) \right)}{\sum_y \mu^I(y \mid s) \exp\left( \tau(Q(s, y) - V(s)) \right)},$$

where $V(s)$ appears in both the numerator and denominator and thus cancels out. This simplifies the expression and leads to:

$$g^*(s' \mid s) = g^{**}(s' \mid s),$$

indicating the equivalence between the advantage-weighted behavior cloning and the $Q$-weighted behavior cloning formulations.

$\square$

## B.7  Proof of Proposition 6

**Proposition.** *The objective $F(\pi)$ is lower-bounded by the following surrogate function $\widetilde{F}(\pi)$, up to an additive constant:* $\widetilde{F}(\pi) = \mathbb{E}_{(s,s') \sim d^I} \left[ \exp\left( \tau Q(s, s') \right) \sum_a \mathcal{I}(a \mid s, s') \log \pi(a \mid s) \right].$

*Proof.* We write the objective function as:

$$F(\pi) = \mathbb{E}_{(s,s') \sim \mathcal{D}^I} \left[ \exp\left( \tau Q(s, s') \right) \log \left( \sum_a \mathcal{T}(s' \mid s, a) \pi(a \mid s) \right) \right].$$

Given that the logarithm function is concave, we apply Jensen's inequality. Define:

$$\Delta(s, s') = \sum_a \mathcal{T}(s' \mid s, a),$$

Then we have:

$$\log \left( \sum_a \mathcal{T}(s' \mid s, a) \pi(a \mid s) \right) = \log \left( \sum_a \frac{\mathcal{T}(s' \mid s, a)}{\Delta(s, s')} \pi(a \mid s) \right) + \log \Delta(s, s') \tag{23}$$

$$\geq \sum_a \frac{\mathcal{T}(s' \mid s, a)}{\Delta(s, s')} \log \pi(a \mid s) + \log \Delta(s, s') \tag{24}$$

$$= \sum_a \mathcal{I}(a \mid s, s') \log \pi(a \mid s) + \log \Delta(s, s'). \tag{25}$$

Substituting this back into the original objective yields the lower bound:

$$F(\pi) \geq \mathbb{E}_{(s,s') \sim \mathcal{D}^I} \left[ \exp\left(\tau Q(s, s')\right) \sum_a \mathcal{I}(a \mid s, s') \log \pi(a \mid s) \right] + \mathbb{E}_{(s,s') \sim \mathcal{D}^I} \left[ \exp\left(\tau Q(s, s')\right) \log \Delta(s, s') \right].$$

The second term is independent of $\pi$ and can be treated as a constant during training. Therefore, we can optimize the surrogate lower bound:

$$\widetilde{F}(\pi) = \mathbb{E}_{(s,s') \sim \mathcal{D}^I} \left[ \exp\left(\tau Q(s, s')\right) \sum_a \mathcal{I}(a \mid s, s') \log \pi(a \mid s) \right].$$

$\square$

## B.8 Complete Derivation of $L(Q, V)$ using Pearson $\chi^2$ divergence

We recall that the objective function $L(Q, V)$ has the following form:

$$\min_Q L(Q, V) = \alpha(1 - \gamma) \mathbb{E}_{(s,s') \sim d_0} [Q(s, s')] + \alpha \mathbb{E}_{(s,s') \sim d^E} [f^* (\gamma V(s') - Q(s, s'))]$$

$$+ (1 - \alpha) \mathbb{E}_{(s,s') \sim d^I} \left[ \widetilde{f^*} (\gamma V(s') - Q(s, s')) \right] \tag{26}$$

where $f^*$ is the convex conjugate of divergence function $f$ and $\widetilde{f^*}(x) = f^*(x) - x$. Under Pearson $\chi^2$ divergence, its convex conjugate $f^*(x) = \frac{x^2}{4} + x$ and the associated $\widetilde{f^*}(x) = \frac{x^2}{4}$. The objective (26) with Pearson $\chi^2$ divergence becomes:

$$\min_Q L(Q, V) = \alpha(1 - \gamma) \mathbb{E}_{(s,s') \sim d_0} [Q(s, s')] + \alpha \mathbb{E}_{(s,s') \sim d^E} [\gamma V(s') - Q(s, s')]$$

$$+ \frac{\alpha}{4} \mathbb{E}_{(s,s') \sim d^E} \left[ (\gamma V(s') - Q(s, s'))^2 \right] + \frac{1 - \alpha}{4} \mathbb{E}_{(s,s') \sim d^I} \left[ (\gamma V(s') - Q(s, s'))^2 \right] \tag{27}$$

$$\Leftrightarrow \min_Q L(Q, V) = (1 - \gamma) \mathbb{E}_{(s,s') \sim d_0} [Q(s, s')] + \mathbb{E}_{(s,s') \sim d^E} [\gamma V(s') - Q(s, s')]$$

$$+ \frac{1}{4\alpha} \alpha \mathbb{E}_{(s,s') \sim d^E} \left[ (\gamma V(s') - Q(s, s'))^2 \right] + \frac{1}{4\alpha} (1 - \alpha) \mathbb{E}_{(s,s') \sim d^I} \left[ (\gamma V(s') - Q(s, s'))^2 \right] \tag{28}$$

$$\Leftrightarrow \min_Q L(Q, V) = (1 - \gamma) \mathbb{E}_{(s,s') \sim d_0} [Q(s, s')] + \mathbb{E}_{s \sim d^E} [\gamma V(s)] - \mathbb{E}_{(s,s') \sim d^E} [Q(s, s')]$$

$$+ \frac{1}{4\alpha} \mathbb{E}_{s,s' \sim d^{E,I}_{mix}} \left[ (\gamma V(s') - Q(s, s'))^2 \right] \tag{29}$$

## B.9 Complete Derivation of $\widetilde{L}(Q, V)$ for bounded Q-learning

The operator $\min_Q -\alpha \mathbb{E}_{(s,s') \sim \mathcal{D}^E} [Q(s, s')]$ in (29) which effectively encourages maximizing the $Q$-values of expert transitions can lead to *unbounded* growth in $Q$, potentially resulting in learning instability. To address this issue, we adapt a technique from [2] that bounds the expert $Q$-values. First,

looking at the objective 28, Let's define $r_Q^E(s, s') = Q(s, s') - \gamma V(s') \ \forall s, s' \sim d^E$. The training objective becomes:

$$\min_Q L(Q, V) = (1 - \gamma) \mathbb{E}_{(s,s') \sim d_0} [Q(s, s')] + \mathbb{E}_{(s,s') \sim d^E} \left[-r_Q^E(s, s')\right]$$

$$+ \frac{1}{4\alpha} \alpha \mathbb{E}_{(s,s') \sim d^E} \left[r_Q^E(s, s')^2\right] + \frac{1}{4\alpha}(1 - \alpha) \mathbb{E}_{(s,s') \sim d^I} \left[(\gamma V(s') - Q(s, s'))^2\right]$$

$$\Leftrightarrow \min_Q L(Q, V) = (1 - \gamma) \mathbb{E}_{d_0(s,s')} \left[Q(s, s')\right] + \frac{1 - \alpha}{4\alpha} \mathbb{E}_{s,s' \sim d^I} [(\gamma V(s') - Q(s, s'))^2]$$

$$+ \left[ \mathbb{E}_{s,s' \sim d^E} [-r_Q^E(s, s')] + \frac{1}{4} \mathbb{E}_{s,s' \sim d^E} [r_Q^E(s, s')^2] \right]$$

$$\Leftrightarrow \min_Q L(Q, V) = (1 - \gamma) \mathbb{E}_{d_0(s,s')} \left[Q(s, s')\right] + \frac{1 - \alpha}{4\alpha} \mathbb{E}_{s,s' \sim d^I} [(\gamma V(s') - Q(s, s'))^2]$$

$$+ \frac{1}{4} \left[ \mathbb{E}_{s,s' \sim d^E} [-4 r_Q^E(s, s')] + \mathbb{E}_{s,s' \sim d^E} [r_Q^E(s, s')^2] + 4 \right] - 1$$

$$\Leftrightarrow \min_Q L(Q, V) = \min_{Q(s,s')} (1 - \gamma) \mathbb{E}_{d_0(s,s')} \left[Q(s, s')\right] + \frac{1 - \alpha}{4\alpha} \mathbb{E}_{s,s' \sim d^I} [(\gamma V(s') - Q(s, s'))^2]$$

$$+ \frac{1}{4} \mathbb{E}_{s,s' \sim d^E} [(r_Q^E(s, s') - 2)^2] \tag{30}$$

$$\Leftrightarrow \min_Q L(Q, V) = (1 - \gamma) \mathbb{E}_{d_0(s,s')} \left[Q(s, s')\right] + \frac{1 - \alpha}{4\alpha} \mathbb{E}_{s,s' \sim d^I} [(\gamma Q(s', g) - Q(s, s'))^2]$$

$$+ \frac{1}{4} \mathbb{E}_{s,s' \sim d^E} [(Q(s, s') - (\gamma V(s') + 2))^2] \tag{31}$$

Following [2], the minimum of the third term in (30) is reached when $r_Q^E(s, s') = 2$. This will lead to $Q(s, s') = \sum_{t=0}^{\infty} \gamma^t 2 = \frac{2}{1-\gamma} \ \forall s, s' \sim d^E$. Therefore, we can replace the target $\gamma V(s') + 2$ in (31) with fixed target $\frac{2}{1-\gamma}$ to have the following modified objective with bounded $Q$.

$$\min_Q \widetilde{L}(Q, V) = (1 - \gamma) \mathbb{E}_{d_0(s,s')} \left[Q(s, s')\right] + \frac{1 - \alpha}{4\alpha} \mathbb{E}_{s,s' \sim d^I} \left[(\gamma V(s') - Q(s, s'))^2\right]$$

$$+ \frac{1}{4} \mathbb{E}_{s,s' \sim d^E} \left[ \left(Q(s, s') - \frac{2}{1-\gamma}\right)^2 \right]$$

## C   Experimental and Implementation Details

Our method is implemented in JAX version 0.5.3 (with CUDA 12 capabilities). We conduct our experiments using a computing cluster with 8 NVIDIA RTX 3090 GPUs. For each IOSTOM run, five distinct training seeds are processed simultaneously on a shared hardware set comprising a single GPU, 32 CPU cores, and 128 GB of RAM. This parallel execution on shared resources enables the completion of 1 million training steps for all five seeds in about 60-90 minutes.

### C.1   Mujoco tasks

We use the same offline LfO benchmark from DILO [38], which utilizes datasets derived from the D4RL [7] framework, and tests on Mujoco environments. The state-only expert dataset in all tasks includes only one expert trajectory. In terms of locomotion tasks, suboptimal datasets, labeled 'random+expert', 'random+few-expert', 'medium+expert', and 'medium+few-expert', are generated by mixing expert trajectories with lower-quality trajectories from D4RL's 'random-v2' and 'medium-v2' datasets, respectively. The 'random+expert' and 'medium+expert' datasets combine 200 expert trajectories with roughly 1 million transitions from the corresponding 'random-v2' or 'medium-v2' dataset. The 'x+few-expert' variants are similar but incorporate only 30 expert trajectories. In manipulation environments, all suboptimal 'x+expert' datasets are formed using 30 expert trajectories mixed with the complete 'x' D4RL dataset. We also use '-v0' variant of D4RL datasets for all manipulation tasks. Table 4 gives an overview about our LfO Mujoco tasks.

| Task | State Dim | Action Dim | Horizon | Suboptimal Dataset | Data Points |
|------|-----------|------------|---------|--------------------|-------------|
| Hopper | 11 | 3 | 1000 | random+expert | 1e6 random transitions + 200 expert trajectories |
| | | | | medium+expert | 1e6 medium transitions + 200 expert trajectories |
| | | | | random+few-expert | 1e6 random transitions + 30 expert trajectories |
| | | | | medium+few-expert | 1e6 medium transitions + 30 expert trajectories |
| Walker2d | 17 | 6 | 1000 | random+expert | 1e6 random transitions + 200 expert trajectories |
| | | | | medium+expert | 1e6 medium transitions + 200 expert trajectories |
| | | | | random+few-expert | 1e6 random transitions + 30 expert trajectories |
| | | | | medium+few-expert | 1e6 medium transitions + 30 expert trajectories |
| Halfcheetah | 17 | 6 | 1000 | random+expert | 1e6 random transitions + 200 expert trajectories |
| | | | | medium+expert | 1e6 medium transitions + 200 expert trajectories |
| | | | | random+few-expert | 1e6 random transitions + 30 expert trajectories |
| | | | | medium+few-expert | 1e6 medium transitions + 30 expert trajectories |
| Ant | 27 | 8 | 1000 | random+expert | 1e6 random transitions + 200 expert trajectories |
| | | | | medium+expert | 1e6 medium transitions + 200 expert trajectories |
| | | | | random+few-expert | 1e6 random transitions + 30 expert trajectories |
| | | | | medium+few-expert | 1e6 medium transitions + 30 expert trajectories |
| Pen | 45 | 24 | 100 | cloned+expert | 5e6 cloned transitions + 30 expert trajectories |
| | | | | human+expert | 5000 human transitions + 30 expert trajectories |
| Door | 39 | 28 | 200 | cloned+expert | 1e6 cloned transitions + 30 expert trajectories |
| | | | | human+expert | 6729 human transitions + 30 expert trajectories |
| Hammer | 46 | 26 | 200 | cloned+expert | 1e6 cloned transitions + 30 expert trajectories |
| | | | | human+expert | 11310 human transitions + 30 expert trajectories |
| Kitchen | 59 | 9 | 280 | partial+expert | 136950 partial transitions + 1 expert trajectories |
| | | | | mixed+expert | 136950 mixed transitions + 1 expert trajectories |

Table 4: Overview of D4RL tasks and their repsective suboptimal dataset we use in LfO setting

## C.2 Maritime Navigation task

The Maritime Navigation task was created using historical data from a hotspot in the Singapore Strait, following the AIS-driven simulation paradigm adopted in recent maritime traffic simulator **ShipNaviSim** [32]. We selected the area with the highest traffic density and collision risk—where numerous ships cross paths, as shown in Figure 3—as our planning region. We collect large amount of historical navigation data ($\sim 2$ years) of vessels operating in this hotspot region recorded in the Automatic Identification System (AIS) from MarineTraffic[1]. The AIS data of each vessel contains two types of information: static and dynamic. The static data contains some information like width, length, type, and ID of vessel. Other vessel movement information like latitude, longitude, speed, heading and course-over-ground are included in dynamic data. To generate trajectory data, we selected tankers and cargo vessels as they represent the riskiest class due to their larger size (200-300 meters) and lower navigational agility. All trajectories were then interpolated at 10-second intervals. The final dataset comprises approximately 125,000 trajectories, totaling around 14 million environment transitions. The average trajectory length in dataset is around 100-150. We used 80% of the data for training and reserved the remaining 20% for evaluation.

The **observation space** is defined from the perspective of the ego agent (the vessel being controlled). At any given time, the agent observes a historical sequence of its own trajectory and those of the 10 closest nearby ships (each over a configurable number of past steps). For the ego agent and nearby ships, and for each historical point, the available features include the $x$ and $y$ coordinates, speed $v$, and heading angle $h$. Additionally, the agent observes its goal location. Observing past states and nearby ship information helps capture multi-ship interactions and provides context for decision-making. For simplicity, all algorithms used the same neural network architecture to process the observation space. We did not use any complex structures; instead, we flattened the observation space and provided it as input to the neural network.

The **action space** is modeled as a straightforward, 3-dimensional continuous space. An action is defined as $\langle d_x, d_y, d_h \rangle$, representing the changes in the $x$ and $y$ coordinates and the change in heading $h$, respectively. The vessel's speed at the next time step is derived from the distance traveled (calculated from $v_{t+1} = \sqrt{d_x^2 + d_y^2}/\delta_T$) divided by the time interval $\delta_T$, which is set to 10 seconds.

---

[1] https://www.marinetraffic.com/

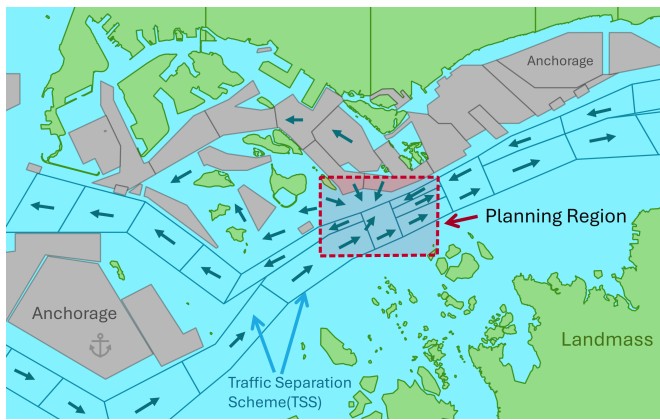

Figure 3: The red region is used as the environment area. The gray areas indicate anchorage zones, the green areas represent landmasses, and the arrows and regions with dark blue borders represent the Maritime Traffic Separation Scheme (TSS). The high density of crossing points in the red area makes it a more challenging region for navigation, providing a suitable setting for testing advanced planning techniques.

This is also known as a delta action space [12] and can be used for any moving object. Because action in our environment represents the difference between some state features of current and next timestep, we can have a simple Inverse Kinematics Model computing this difference to infer action between two consecutive states of state-only trajectories in datasets.

Following **vessel-specific metrics** introduced in **ShipNaviSim** [32] are used to evaluate navigation policies, comparing learned agent behavior to human expert data.

**Goal-Conditioned ADE (GC-ADE)** measures the average displacement between the learned policy's trajectory and the original historical trajectory in the 2D plane. Given $\tau_m$ of length $T_m$ and $\tau_p$ of length $T_p$, GC-ADE computes the error over the minimum of the two lengths.

$$\text{GC-ADE} = \frac{1}{\min(T_m, T_p)} \sqrt{\sum_{t=1}^{\min(T_m, T_p)} (x_t^m - x_t^p)^2 + (y_t^m - y_t^p)^2}$$

**Goal Rate** is the percentage of times the ego agent successfully reaches its designated goal location. Success is defined as coming within a radius of 200 meters of the goal.

**Near Miss Count** represents the average number of time-steps per episode during which the ego agent approaches another vessel within a distance of 3 cable lengths (555 meters), which is considered a near-miss by domain experts. The 'near-miss' metric is interpreted broadly as a proxy for high traffic density and increased potential for navigation risk; it does not always imply that vessels in 'near-miss' situation were about to collide.

### C.3   Architecture and Hyperparameters

Our implementation builds upon the official implementations of ReCOIL [39] and XQL [9]. We keep most of their parameters and network settings as shown in Table 5. We also add Layer Normalization [3] in V-function network to improve training stability as suggested in XQL. The regularization $\beta$ was tuned by searching over [3, 5, 7, 10, 15, 20]. For locomotion tasks, we set $\beta = 20$ for standard LfO setting, and $\beta = 15$ for subsampled setting. In terms of manipulation tasks, $\beta = 10$ works best in most cases except 'pen-cloned' setting where $\beta$ is set to 3. The policy temperature $\tau$ is often set to 3 in previous works [22, 39]. However, we find that this value results in very bad performance for IOSTOM because we do not use advantage for updating policy. We tune $\tau$ via via hyper-parameter sweeps over [0.01, 0.04, 0.08, 0.1, 0.2]. $\tau = 0.04$ is the best-performing hyperparameter in most tasks except for the 'human' Adroit and 'mixed' Franka Kitchen manipulation tasks, where $\tau = 0.01$ was used. For maritime navigation task, we set $\beta = 20$ and $\tau = 0.04$ which is similar to LfO setting of locomotion tasks.

| Type | Hyperparameter | Value |
|---|---|---|
| Actor | Network Size | [256, 256] |
| | Activation Function | ReLU |
| | Learning Rate | 3e-4 |
| | Weight Decay | 1e-3 |
| | Training Length | 1M steps |
| | Batch Size | 512 |
| | Optimizer | Adam |
| | Dropout Rate | 0.1 |
| | LR decay schedule | cosine |
| Critic | Network Size | [256, 256] |
| | Activation Function | ReLU |
| | Learning Rate | 3e-4 |
| | Training Length | 1M steps |
| | Batch Size | 512 |
| | Optimizer | Adam |
| | Mixture Ratio $\alpha$ | 0.5 |
| | Polyak Update Rate $\lambda$ | 0.005 |
| | Discount Factor $\gamma$ | 0.99 |

Table 5: Hyperparameters of IOSTOM

## C.4 Baseslines

To evaluate the performance of our approach, we conduct comparative evaluations against three established state-of-the-art (SOTA) techniques: SMODICE [27], PW-DICE [48], and DILO [38]. The SMODICE and PW-DICE algorithms both operate by training a discriminator to guide the learned policy. Their fundamental difference lies in the divergence measure employed: SMODICE seeks to minimize the KL-divergence between the state occupancies of the learner and the expert, while PW-DICE alternatively uses the Wasserstein distance for this alignment. DILO offers a distinct, more recent SOTA paradigm for LfO, notable for its discriminator-free learning process. For all comparative methods, we utilize the publicly accessible codebases provided by their authors. To ensure fair comparisons, we use the hyperparameter settings recommended in their original publications or the default configurations within their code. The only exception is DILO where we can not reproduce consistent results as reported in the paper using their default parameters. After some tuning effort, we find that using Layer Normalization [3] can help to improve DILO performance. However, the training still diverges in some tasks as shown in Figures 4 and 5

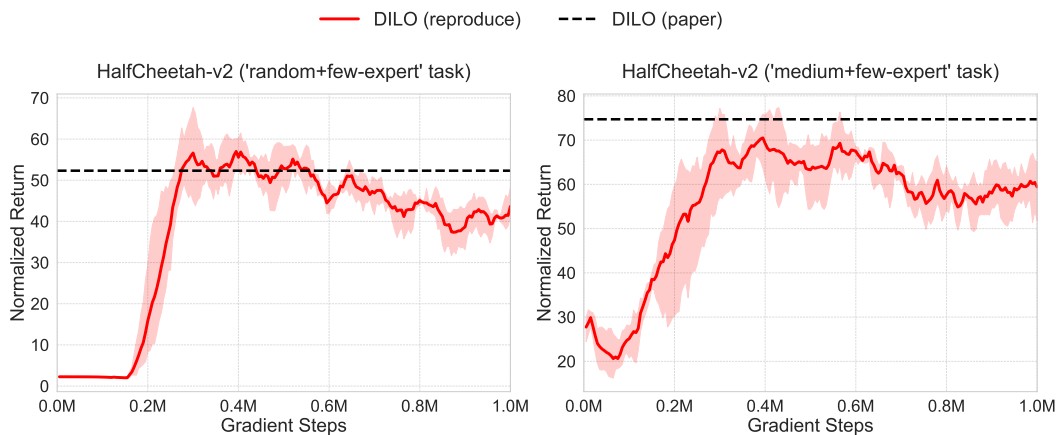

Figure 4: Training divergence of DILO on locomotion tasks

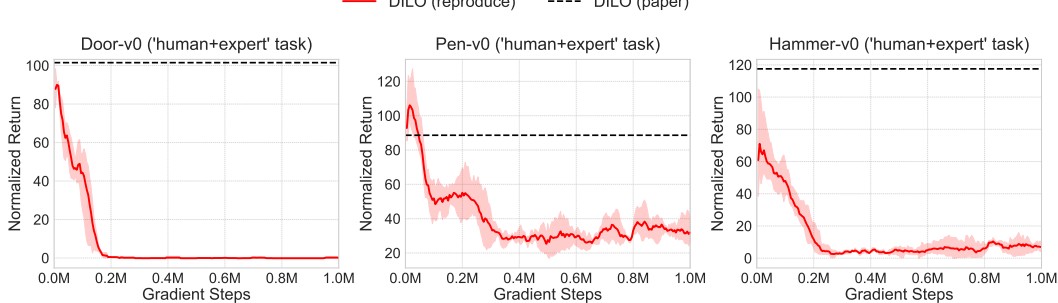

Figure 5: Training divergence of DILO on manipulation tasks

# D  Additional Experiments

## D.1  Comparison with LobsDICE

| Suboptimal Dataset | Env | PW-DICE | LobsDICE | IOSTOM |
|---|---|---|---|---|
| random+expert | hopper
halfcheetah
walker2d
ant | 108.09 $_{\pm 2.39}$
86.11 $_{\pm 4.39}$
107.48 $_{\pm 0.53}$
126.89 $_{\pm 1.17}$ | 99.64 $_{\pm 5.01}$
80.76 $_{\pm 3.17}$
107.54 $_{\pm 0.13}$
122.65 $_{\pm 0.72}$ | **109.32** $_{\pm 1.08}$
**93.02** $_{\pm 0.40}$
**107.98** $_{\pm 0.20}$
**128.19** $_{\pm 1.52}$ |
| random+few-expert | hopper
halfcheetah
walker2d
ant | 75.04 $_{\pm 14.21}$
4.02 $_{\pm 1.74}$
36.11 $_{\pm 9.19}$
99.90 $_{\pm 2.59}$ | 74.25 $_{\pm 11.23}$
11.61 $_{\pm 5.66}$
101.29 $_{\pm 5.53}$
93.84 $_{\pm 7.51}$ | **107.28** $_{\pm 3.92}$
**88.77** $_{\pm 1.26}$
**108.40** $_{\pm 0.21}$
**120.09** $_{\pm 5.17}$ |
| medium+expert | hopper
halfcheetah
walker2d
ant | 65.99 $_{\pm 8.05}$
58.74 $_{\pm 1.84}$
105.41 $_{\pm 0.33}$
108.14 $_{\pm 1.90}$ | 74.43 $_{\pm 5.28}$
71.09 $_{\pm 3.76}$
103.34 $_{\pm 2.11}$
114.96 $_{\pm 4.05}$ | **110.20** $_{\pm 0.51}$
**93.12** $_{\pm 0.32}$
**108.12** $_{\pm 0.13}$
**124.72** $_{\pm 3.49}$ |
| medium+few-expert | hopper
halfcheetah
walker2d
ant | 57.24 $_{\pm 3.03}$
27.85 $_{\pm 6.03}$
75.22 $_{\pm 7.05}$
90.34 $_{\pm 2.56}$ | 67.00 $_{\pm 6.43}$
44.76 $_{\pm 3.94}$
95.39 $_{\pm 5.22}$
95.33 $_{\pm 1.08}$ | **108.96** $_{\pm 1.33}$
**89.47** $_{\pm 0.82}$
**108.15** $_{\pm 0.43}$
**120.36** $_{\pm 1.25}$ |
| cloned+expert | pen
door
hammer | 23.39 $_{\pm 4.56}$
0.07 $_{\pm 0.14}$
1.29 $_{\pm 0.12}$ | 29.83 $_{\pm 4.59}$
0.02 $_{\pm 0.00}$
0.55 $_{\pm 0.19}$ | **82.77** $_{\pm 4.84}$
**102.77** $_{\pm 0.96}$
**94.59** $_{\pm 9.39}$ |
| human+expert | pen
door
hammer | -2.56 $_{\pm 1.30}$
0.15 $_{\pm 0.02}$
2.02 $_{\pm 0.77}$ | 42.09 $_{\pm 5.08}$
10.98 $_{\pm 9.71}$
17.06 $_{\pm 13.44}$ | **95.77** $_{\pm 8.91}$
**100.77** $_{\pm 1.68}$
**93.34** $_{\pm 7.41}$ |
| partial+expert | kitchen | 12.33 $_{\pm 5.38}$ | 40.33 $_{\pm 2.24}$ | **58.95** $_{\pm 2.27}$ |
| mixed+expert | kitchen | 7.50 $_{\pm 4.16}$ | 45.67 $_{\pm 2.75}$ | **46.45** $_{\pm 0.84}$ |

Table 6:  Average normalized return over last 10 evaluations of IOSTOM against LobsDICE and PW-DICE on the D4RL suboptimal datasets with 1 expert trajectory. The mean and std are obtained over 5 random seeds. LfO methods with avg. perf within the std-dev of the top performing LfO approach is in **bold**.

To further strengthen our empirical study, we additionally include a comparison against LobsDICE [18], alongside PW-DICE and our method (IOSTOM), under the same experimental settings. Although previous work [48] has suggested that PW-DICE generally outperforms LobsDICE on the

D4RL "random+expert" benchmark, we perform a direct comparison here for completeness and to ensure a fair evaluation across representative DICE-based baselines.

The corresponding results are reported in Table 6. We observe that IOSTOM consistently and substantially outperforms LobsDICE across all tasks and dataset regimes. As is common among DICE-based methods, LobsDICE exhibits degraded performance in the "few-expert" and manipulation tasks, whereas IOSTOM remains robust. Our findings also confirm that PW-DICE surpasses LobsDICE in the "random+expert" setting, but PW-DICE becomes unstable and underperforms in several other configurations, illustrating the general sensitivity of discriminator-based approaches. In contrast, IOSTOM achieves both stronger performance and greater robustness, reinforcing the benefits of its stable, discriminator-free formulation.

## D.2 Sensitivity to Subsampling of Low-Quality Data

| Suboptimal Dataset | Env | IOSTOM(full) | IOSTOM(sub5) | IOSTOM(sub20) |
|---|---|---|---|---|
| random+expert | hopper | $109.32_{\pm1.08}$ | $110.24_{\pm0.63}$ | $110.66_{\pm0.16}$ |
| | halfcheetah | $93.02_{\pm0.40}$ | $90.81_{\pm2.10}$ | $\mathbf{85.59}_{\pm5.09}$ |
| | walker2d | $107.98_{\pm0.20}$ | $107.91_{\pm0.16}$ | $107.53_{\pm0.11}$ |
| | ant | $128.19_{\pm1.52}$ | $127.26_{\pm2.48}$ | $127.30_{\pm1.37}$ |
| random+few-expert | hopper | $107.28_{\pm3.92}$ | $108.99_{\pm0.76}$ | $109.34_{\pm1.62}$ |
| | halfcheetah | $88.77_{\pm1.26}$ | $86.62_{\pm1.98}$ | $\mathbf{80.49}_{\pm6.41}$ |
| | walker2d | $108.40_{\pm0.21}$ | $108.13_{\pm0.29}$ | $107.78_{\pm0.22}$ |
| | ant | $120.09_{\pm5.17}$ | $119.63_{\pm2.40}$ | $\mathbf{105.98}_{\pm7.03}$ |
| medium+expert | hopper | $110.20_{\pm0.51}$ | $109.89_{\pm0.22}$ | $109.58_{\pm1.18}$ |
| | halfcheetah | $93.12_{\pm0.32}$ | $91.07_{\pm1.72}$ | $\mathbf{76.30}_{\pm13.56}$ |
| | walker2d | $108.12_{\pm0.13}$ | $108.31_{\pm0.21}$ | $108.03_{\pm0.15}$ |
| | ant | $124.72_{\pm3.49}$ | $127.71_{\pm2.72}$ | $128.61_{\pm0.71}$ |
| medium+few-expert | hopper | $108.96_{\pm1.33}$ | $109.68_{\pm0.56}$ | $108.20_{\pm0.83}$ |
| | halfcheetah | $89.47_{\pm0.82}$ | $90.20_{\pm1.94}$ | $\mathbf{77.65}_{\pm11.48}$ |
| | walker2d | $108.15_{\pm0.43}$ | $107.88_{\pm0.83}$ | $107.39_{\pm0.92}$ |
| | ant | $120.36_{\pm1.25}$ | $118.23_{\pm5.04}$ | $\mathbf{102.14}_{\pm7.26}$ |

Table 7: Average normalized return over the last 10 evaluations of IOSTOM under different subsampling rates of low-quality data (sub5 and sub20) on the D4RL suboptimal datasets with 1 expert trajectory. Performance drops exceeding 5% relative to IOSTOM(full) are shown in **bold**.

To assess the sensitivity of our method to the amount of available low-quality data, we conducted additional experiments where the random (or medium-quality) portion of the dataset was subsampled to only 20% and 5% of its original size. These two variants are denoted as *IOSTOM(sub5)* and *IOSTOM(sub20)*, while the original version is referred to as *IOSTOM(full)*. The corresponding results are summarized in Table 7. Performance drops exceeding 5% relative to IOSTOM(full) are highlighted in **bold**.

When reducing the low-quality data to 20% (IOSTOM(sub5)), our method exhibits strong robustness: across all tasks and datasets, performance remains very close to IOSTOM(full), with no significant degradation observed. However, when the low-quality portion is aggressively reduced to just 5% (IOSTOM(sub20)), we observe a more noticeable performance decline — up to 18% on some tasks, particularly in halfcheetah and ant. Nonetheless, IOSTOM still achieves robust performance on 10 out of 16 tasks, even under this extremely limited data regime.

## D.3 Comparison with other variants of IOSTOM

To validate our algorithmic designs, we compare IOSTOM with other variants: IOSTOM-IDM (Using Inverse Dynamics Model), IOSTOM-Adv (Using advantage instead of Q to update policy), and IOSTOM-IQL (Using Implicit Q Learning [22] objective to train V-function network). Table 8 shows these comparison results. Overall, IOSTOM has the best performance on 17/24 tasks and consistently produces high-quality results compared to other variants. IOSTOM-IQL is the second

| Suboptimal Dataset | Env | IOSTOM-ADV | IOSTOM-IDM | IOSTOM-IQL | IOSTOM | Expert |
|---|---|---|---|---|---|---|
| random+expert | hopper | $55.93_{\pm6.67}$ | $97.06_{\pm4.57}$ | $\mathbf{109.74}_{\pm0.26}$ | $\mathbf{109.32}_{\pm1.08}$ | 111.33 |
| | halfcheetah | $6.43_{\pm2.51}$ | $80.53_{\pm3.40}$ | $\mathbf{92.82}_{\pm0.71}$ | $\mathbf{93.02}_{\pm0.40}$ | 88.83 |
| | walker2d | $104.20_{\pm4.07}$ | $79.70_{\pm29.28}$ | $\mathbf{108.43}_{\pm0.14}$ | $107.98_{\pm0.20}$ | 106.92 |
| | ant | $120.13_{\pm3.65}$ | $\mathbf{128.99}_{\pm3.11}$ | $\mathbf{128.71}_{\pm3.73}$ | $\mathbf{128.19}_{\pm1.52}$ | 130.75 |
| random+few-expert | hopper | $20.47_{\pm10.26}$ | $50.63_{\pm32.03}$ | $106.59_{\pm3.05}$ | $\mathbf{107.28}_{\pm3.92}$ | 111.33 |
| | halfcheetah | $2.13_{\pm0.11}$ | $68.79_{\pm4.26}$ | $86.44_{\pm1.55}$ | $\mathbf{88.77}_{\pm1.26}$ | 88.83 |
| | walker2d | $8.38_{\pm2.84}$ | $69.11_{\pm23.28}$ | $108.37_{\pm0.05}$ | $\mathbf{108.40}_{\pm0.21}$ | 106.92 |
| | ant | $41.18_{\pm12.90}$ | $\mathbf{123.86}_{\pm1.97}$ | $125.15_{\pm4.50}$ | $120.09_{\pm5.17}$ | 130.75 |
| medium+expert | hopper | $53.46_{\pm13.16}$ | $97.62_{\pm7.42}$ | $\mathbf{110.71}_{\pm0.35}$ | $110.20_{\pm0.51}$ | 111.33 |
| | halfcheetah | $49.00_{\pm3.44}$ | $85.21_{\pm1.86}$ | $91.45_{\pm1.49}$ | $\mathbf{93.12}_{\pm0.32}$ | 88.83 |
| | walker2d | $85.96_{\pm21.33}$ | $94.90_{\pm29.34}$ | $\mathbf{108.45}_{\pm0.17}$ | $108.12_{\pm0.13}$ | 106.92 |
| | ant | $118.74_{\pm4.55}$ | $\mathbf{128.32}_{\pm0.64}$ | $124.66_{\pm4.62}$ | $124.72_{\pm3.49}$ | 130.75 |
| medium few-expert | hopper | $42.79_{\pm2.36}$ | $68.32_{\pm17.89}$ | $106.02_{\pm3.31}$ | $\mathbf{108.96}_{\pm1.33}$ | 111.33 |
| | halfcheetah | $42.31_{\pm0.60}$ | $76.48_{\pm5.57}$ | $78.18_{\pm2.24}$ | $\mathbf{89.47}_{\pm0.82}$ | 88.83 |
| | walker2d | $74.16_{\pm2.14}$ | $107.89_{\pm0.28}$ | $\mathbf{108.38}_{\pm0.16}$ | $108.15_{\pm0.43}$ | 106.92 |
| | ant | $99.89_{\pm1.92}$ | $\mathbf{121.80}_{\pm1.78}$ | $\mathbf{121.64}_{\pm2.35}$ | $120.36_{\pm1.25}$ | 130.75 |
| cloned+expert | pen | $41.65_{\pm5.42}$ | $63.39_{\pm11.46}$ | $10.54_{\pm1.51}$ | $\mathbf{82.77}_{\pm4.84}$ | 106.42 |
| | door | $13.87_{\pm8.26}$ | $18.68_{\pm12.44}$ | $32.25_{\pm13.03}$ | $\mathbf{102.77}_{\pm0.96}$ | 103.94 |
| | hammer | $11.77_{\pm16.66}$ | $47.83_{\pm8.43}$ | $57.04_{\pm6.34}$ | $\mathbf{94.59}_{\pm9.39}$ | 125.71 |
| human+expert | pen | $\mathbf{92.73}_{\pm3.73}$ | $81.72_{\pm5.13}$ | $\mathbf{95.26}_{\pm10.16}$ | $\mathbf{95.77}_{\pm8.91}$ | 106.42 |
| | door | $95.08_{\pm1.90}$ | $78.50_{\pm20.55}$ | $\mathbf{99.47}_{\pm3.67}$ | $\mathbf{100.77}_{\pm1.68}$ | 103.94 |
| | hammer | $\mathbf{88.23}_{\pm5.72}$ | $82.12_{\pm18.51}$ | $68.32_{\pm13.66}$ | $\mathbf{93.34}_{\pm7.41}$ | 125.71 |
| partial+expert | kitchen | $56.08_{\pm0.29}$ | $66.30_{\pm5.70}$ | $57.75_{\pm2.00}$ | $58.95_{\pm2.27}$ | 75.0 |
| mixed+expert | kitchen | $\mathbf{49.42}_{\pm0.58}$ | $28.95_{\pm10.62}$ | $47.92_{\pm1.23}$ | $46.45_{\pm0.84}$ | 75.0 |

Table 8: Average normalized return over last 10 evaluations of IOSTOM against other variants on the D4RL suboptimal datasets with 1 expert trajectory. The mean and std are obtained over 5 random seeds. LfO methods with avg. perf within the std-dev of the top performing LfO approach is in **bold**.

.

best method, but its performance is still significantly worse than IOSTOM on 'cloned' tasks. The results in 'few-expert' setting of IOSTOM-IDM is very bad compared to 'expert' setting which clearly shows the weakness of training Inverse Dynamics Model with low-quality data. IOSTOM-ADV has the worst performance in most tasks.

## D.4 $\alpha$ Ablation

| Suboptimal Dataset | Env | $\alpha = 0.1$ | $\alpha = 0.3$ | $\alpha = 0.5$ (our) | $\alpha = 0.7$ | $\alpha = 0.9$ | $gap_{worst}(\%)$ | $gap_{default}(\%)$ |
|---|---|---|---|---|---|---|---|---|
| random+expert | hopper | $109.72_{\pm0.84}$ | $110.21_{\pm0.68}$ | $109.32_{\pm1.08}$ | $109.19_{\pm0.21}$ | $110.29_{\pm0.64}$ | 1.00 | 0.88 |
| | halfcheetah | $\mathbf{93.09}_{\pm0.22}$ | $92.90_{\pm0.31}$ | $93.02_{\pm0.40}$ | $92.99_{\pm0.08}$ | $92.89_{\pm0.18}$ | 0.21 | 0.08 |
| | walker2d | $107.82_{\pm0.23}$ | $107.97_{\pm0.08}$ | $107.98_{\pm0.20}$ | $108.16_{\pm0.17}$ | $\mathbf{108.38}_{\pm0.26}$ | 0.52 | 0.37 |
| | ant | $\mathbf{128.37}_{\pm1.89}$ | $126.67_{\pm2.89}$ | $128.19_{\pm1.52}$ | $125.78_{\pm3.00}$ | $127.30_{\pm1.85}$ | 2.02 | 0.14 |
| random+few-expert | hopper | $107.41_{\pm1.27}$ | $\mathbf{108.59}_{\pm2.04}$ | $107.28_{\pm3.92}$ | $104.75_{\pm3.36}$ | $107.25_{\pm4.11}$ | 3.54 | 1.21 |
| | halfcheetah | $87.29_{\pm0.73}$ | $87.52_{\pm2.11}$ | $\mathbf{88.77}_{\pm1.26}$ | $88.42_{\pm1.00}$ | $86.61_{\pm1.84}$ | 2.43 | 0.00 |
| | walker2d | $108.09_{\pm0.24}$ | $108.21_{\pm0.06}$ | $108.40_{\pm0.21}$ | $\mathbf{108.45}_{\pm0.12}$ | $108.24_{\pm0.14}$ | 0.33 | 0.05 |
| | ant | $\mathbf{125.43}_{\pm2.77}$ | $121.19_{\pm2.09}$ | $120.09_{\pm5.17}$ | $122.18_{\pm1.90}$ | $123.46_{\pm1.17}$ | 4.26 | 4.26 |
| medium+expert | hopper | $109.79_{\pm0.95}$ | $110.44_{\pm0.51}$ | $110.20_{\pm0.51}$ | $110.30_{\pm0.54}$ | $\mathbf{110.61}_{\pm0.07}$ | 0.74 | 0.37 |
| | halfcheetah | $\mathbf{93.16}_{\pm0.19}$ | $92.82_{\pm0.32}$ | $93.12_{\pm0.32}$ | $93.00_{\pm0.23}$ | $92.68_{\pm0.17}$ | 0.52 | 0.04 |
| | walker2d | $107.54_{\pm0.45}$ | $107.96_{\pm0.12}$ | $108.12_{\pm0.13}$ | $107.28_{\pm1.44}$ | $\mathbf{108.22}_{\pm0.27}$ | 0.87 | 0.09 |
| | ant | $\mathbf{129.00}_{\pm0.59}$ | $124.91_{\pm2.56}$ | $124.72_{\pm3.49}$ | $125.52_{\pm2.26}$ | $127.86_{\pm2.20}$ | 3.32 | 3.32 |
| medium+few-expert | hopper | $107.95_{\pm1.72}$ | $\mathbf{110.08}_{\pm1.12}$ | $108.96_{\pm1.33}$ | $106.99_{\pm4.23}$ | $104.34_{\pm5.88}$ | 5.21 | 1.02 |
| | halfcheetah | $87.98_{\pm1.17}$ | $87.79_{\pm1.77}$ | $\mathbf{89.47}_{\pm0.82}$ | $88.66_{\pm1.31}$ | $89.17_{\pm0.60}$ | 1.88 | 0.00 |
| | walker2d | $108.15_{\pm0.31}$ | $108.24_{\pm0.33}$ | $108.15_{\pm0.43}$ | $108.16_{\pm0.22}$ | $\mathbf{108.47}_{\pm0.22}$ | 0.30 | 0.30 |
| | ant | $120.44_{\pm1.57}$ | $119.36_{\pm1.84}$ | $120.36_{\pm1.25}$ | $119.12_{\pm0.58}$ | $\mathbf{122.32}_{\pm0.73}$ | 2.62 | 1.60 |
| cloned+expert | pen | $50.08_{\pm17.30}$ | $72.62_{\pm7.45}$ | $\mathbf{82.77}_{\pm4.84}$ | $73.96_{\pm5.58}$ | $76.37_{\pm6.52}$ | 39.49 | 0.00 |
| | door | $102.68_{\pm0.63}$ | $\mathbf{103.79}_{\pm0.63}$ | $102.77_{\pm0.96}$ | $102.65_{\pm1.37}$ | $103.64_{\pm0.92}$ | 1.10 | 0.98 |
| | hammer | $101.98_{\pm4.88}$ | $\mathbf{110.07}_{\pm7.29}$ | $94.59_{\pm9.39}$ | $105.58_{\pm12.35}$ | $106.81_{\pm2.94}$ | 14.06 | 14.06 |
| human+expert | pen | $95.75_{\pm7.18}$ | $96.68_{\pm4.83}$ | $95.77_{\pm8.91}$ | $\mathbf{98.29}_{\pm5.11}$ | $97.74_{\pm5.11}$ | 2.58 | 2.56 |
| | door | $100.41_{\pm3.85}$ | $100.94_{\pm3.65}$ | $100.77_{\pm1.68}$ | $99.99_{\pm2.31}$ | $\mathbf{101.13}_{\pm1.07}$ | 1.13 | 0.36 |
| | hammer | $95.24_{\pm5.79}$ | $101.71_{\pm4.97}$ | $93.34_{\pm7.41}$ | $101.04_{\pm6.36}$ | $\mathbf{103.79}_{\pm8.80}$ | 10.07 | 10.07 |
| partial+expert | kitchen | $\mathbf{64.00}_{\pm7.66}$ | $61.83_{\pm1.38}$ | $58.95_{\pm2.27}$ | $61.33_{\pm4.13}$ | $60.08_{\pm3.21}$ | 7.89 | 7.89 |
| mixed+expert | kitchen | $45.25_{\pm2.38}$ | $45.42_{\pm4.05}$ | $\mathbf{46.45}_{\pm0.84}$ | $45.08_{\pm1.13}$ | $45.33_{\pm2.01}$ | 2.95 | 0.00 |
| Average | | | | | | | 4.54 | 2.07 |

Table 9: Average normalized return over last 10 evaluations of IOSTOM with different $\alpha$ values on the D4RL suboptimal datasets with 1 expert trajectory. Method with the best avg. perf is in **bold**.

This section presents an ablation study to evaluate the impact of the hyperparameter $\alpha$ on IOSTOM's performance. Table D.4 reports performance for each setting, with the best-performing $\alpha$ for each task highlighted in bold. We also include two additional metrics: $gap_{worst}$, representing the percentage gap between the best and worst $\alpha$, and $gap_{default}$, indicating the gap between the best-performing $\alpha$ and our default setting of $\alpha = 0.5$.

According to the table, the average $gap_{worst}$ value across all tasks is just 4.54%, which is relatively small. This indicates that IOSTOM's performance is not highly sensitive to the choice of $\alpha$. Furthermore, the average performance gap between the task-specific optimal $\alpha$ and our default choice of $\alpha = 0.5$ is even smaller at 2.07%. The $gap_{default}$ value is also under 5% in all but 3 of 24 tasks. This confirms that $\alpha = 0.5$ is a robust and effective hyperparameter choice, consistently providing near-optimal performance.

## D.5 $\beta$ Ablation

| Suboptimal Dataset | Env | $\beta$=3 | $\beta$=5 | $\beta$=10 | $\beta$=15 | $\beta$=20 | Expert |
|---|---|---|---|---|---|---|---|
| random+ expert | hopper | 13.72 $_{\pm 3.59}$ | 5.56 $_{\pm 1.16}$ | 41.39 $_{\pm 39.18}$ | **110.36** $_{\pm 0.46}$ | 109.32 $_{\pm 1.08}$ | 111.33 |
| | halfcheetah | 52.40 $_{\pm 11.29}$ | 92.64 $_{\pm 0.90}$ | 93.10 $_{\pm 0.25}$ | **93.18** $_{\pm 0.35}$ | 93.02 $_{\pm 0.40}$ | 88.83 |
| | walker2d | 1.18 $_{\pm 0.29}$ | 60.97 $_{\pm 39.06}$ | 6.75 $_{\pm 14.23}$ | 107.67 $_{\pm 0.14}$ | **107.98** $_{\pm 0.20}$ | 106.92 |
| | ant | 118.94 $_{\pm 7.69}$ | 126.17 $_{\pm 2.26}$ | 128.02 $_{\pm 3.01}$ | **128.20** $_{\pm 3.52}$ | 128.19 $_{\pm 1.52}$ | 130.75 |
| random+ few-expert | hopper | 10.68 $_{\pm 3.56}$ | 6.10 $_{\pm 0.76}$ | 35.94 $_{\pm 15.77}$ | 102.30 $_{\pm 5.12}$ | **107.28** $_{\pm 3.92}$ | 111.33 |
| | halfcheetah | 2.24 $_{\pm 0.01}$ | 2.20 $_{\pm 0.05}$ | 83.37 $_{\pm 2.08}$ | 85.20 $_{\pm 1.61}$ | **88.77** $_{\pm 1.26}$ | 88.83 |
| | walker2d | 1.20 $_{\pm 0.24}$ | 11.65 $_{\pm 3.33}$ | 29.87 $_{\pm 31.90}$ | 108.21 $_{\pm 0.41}$ | **108.40** $_{\pm 0.21}$ | 106.92 |
| | ant | 45.72 $_{\pm 15.95}$ | 97.42 $_{\pm 14.16}$ | **125.28** $_{\pm 3.05}$ | 122.76 $_{\pm 3.64}$ | 120.09 $_{\pm 5.17}$ | 130.75 |
| medium+ expert | hopper | 31.99 $_{\pm 24.17}$ | 61.48 $_{\pm 19.37}$ | 109.97 $_{\pm 0.42}$ | 110.02 $_{\pm 1.00}$ | **110.20** $_{\pm 0.51}$ | 111.33 |
| | halfcheetah | 43.20 $_{\pm 0.47}$ | 54.63 $_{\pm 3.08}$ | 92.63 $_{\pm 0.21}$ | 92.96 $_{\pm 0.29}$ | **93.12** $_{\pm 0.32}$ | 88.83 |
| | walker2d | 71.70 $_{\pm 4.56}$ | 107.83 $_{\pm 0.75}$ | **108.31** $_{\pm 0.27}$ | 108.28 $_{\pm 0.12}$ | 108.12 $_{\pm 0.13}$ | 106.92 |
| | ant | 98.70 $_{\pm 1.81}$ | 102.01 $_{\pm 2.96}$ | 121.86 $_{\pm 2.49}$ | 124.12 $_{\pm 2.93}$ | **124.72** $_{\pm 3.49}$ | 130.75 |
| medium few-expert | hopper | 46.86 $_{\pm 3.74}$ | 61.88 $_{\pm 22.27}$ | 105.83 $_{\pm 4.07}$ | 106.80 $_{\pm 2.29}$ | **108.96** $_{\pm 1.33}$ | 111.33 |
| | halfcheetah | 42.85 $_{\pm 0.27}$ | 43.17 $_{\pm 0.12}$ | 49.01 $_{\pm 1.15}$ | 83.52 $_{\pm 1.72}$ | **89.47** $_{\pm 0.82}$ | 88.83 |
| | walker2d | 66.58 $_{\pm 1.99}$ | 69.35 $_{\pm 6.58}$ | 108.33 $_{\pm 0.28}$ | **108.46** $_{\pm 0.13}$ | 108.15 $_{\pm 0.43}$ | 106.92 |
| | ant | 92.15 $_{\pm 1.80}$ | 94.62 $_{\pm 3.44}$ | 98.59 $_{\pm 1.67}$ | 111.45 $_{\pm 3.09}$ | **120.36** $_{\pm 1.25}$ | 130.75 |
| cloned+expert | pen | **82.77** $_{\pm 4.84}$ | 56.05 $_{\pm 7.20}$ | 11.30 $_{\pm 2.64}$ | 10.33 $_{\pm 2.27}$ | 10.13 $_{\pm 2.90}$ | 106.42 |
| | door | 40.58 $_{\pm 55.52}$ | 80.99 $_{\pm 45.33}$ | **102.77** $_{\pm 0.96}$ | 100.08 $_{\pm 2.59}$ | 86.06 $_{\pm 6.14}$ | 103.94 |
| | hammer | 88.63 $_{\pm 33.94}$ | 94.88 $_{\pm 16.51}$ | 94.59 $_{\pm 9.39}$ | **100.59** $_{\pm 10.60}$ | 90.27 $_{\pm 18.70}$ | 125.71 |
| human+expert | pen | 99.27 $_{\pm 5.22}$ | **101.27** $_{\pm 6.45}$ | 95.77 $_{\pm 8.91}$ | 96.14 $_{\pm 7.88}$ | 99.96 $_{\pm 13.06}$ | 106.42 |
| | door | 99.90 $_{\pm 1.90}$ | **102.22** $_{\pm 1.42}$ | 100.77 $_{\pm 1.68}$ | 99.43 $_{\pm 2.36}$ | 101.22 $_{\pm 2.95}$ | 103.94 |
| | hammer | 87.11 $_{\pm 16.28}$ | **102.10** $_{\pm 15.65}$ | 93.34 $_{\pm 7.41}$ | 97.37 $_{\pm 15.52}$ | 93.39 $_{\pm 7.70}$ | 125.71 |
| partial+expert | kitchen | 49.80 $_{\pm 14.81}$ | **61.10** $_{\pm 3.24}$ | 57.75 $_{\pm 2.00}$ | 63.00 $_{\pm 3.33}$ | 59.70 $_{\pm 5.28}$ | 75.0 |
| mixed+expert | kitchen | 46.75 $_{\pm 1.63}$ | 45.80 $_{\pm 2.65}$ | **47.92** $_{\pm 1.23}$ | 46.85 $_{\pm 1.80}$ | 45.40 $_{\pm 3.84}$ | 75.0 |

Table 10: Average normalized return over last 10 evaluations of IOSTOM with different $\beta$ values on the D4RL suboptimal datasets with 1 expert trajectory. Method with the best avg. perf is in **bold**.

This section presents an ablation study to evaluate the impact of the hyperparameter $\beta$ on IOSTOM's performance. Table 10 summarizes these results. For locomotion tasks (e.g., Hopper, HalfCheetah, Walker2d, Ant), higher $\beta$ values, typically 15 or 20, generally yield superior scores compared to lower values such as 3 or 5. Conversely, for manipulation tasks (e.g., Pen, Door, Hammer, Kitchen), optimal performance is often achieved with $\beta$ values of 5 or 10. However, the performance differences across various $\beta$ settings for these tasks are less pronounced. The only exception is the 'pen' environment within the 'cloned+expert' dataset, where decreasing $\beta$ leads to improved results, with $\beta = 3$ achieving the highest score.

