# OpenReview forum: "IOSTOM: Offline Imitation Learning from Observations via State Transition Occupancy Matching"
_NeurIPS.cc/2025/Conference — NeurIPS 2025 poster_

### Official Review · Reviewer_y8cD · 2025-06-28

**Clarity:** 2
**Significance:** 2
**Originality:** 3
**Rating:** 3
**Confidence:** 4

**Summary:**

The paper presents imitation learning from observation via state transition occupancy matching (IOSTOM), a new offline LfO algorithm that excludes actions entirely from the training objective. The proposed algorithm learn an implicit policy that models transition probabilities between states, rather than leveraging action information of non-expert demonstrations. In the experiments, the authors demonstrate that IOSTOM outperforms other baseline algorithms on diverse sets of environments and tasks from D4RL benchmark.

**Questions:**

- What is the main reason for considering mixed state visitation disruption instead of starting from directly matching the objective with $d^E$? (What is the main problem if we derive it in the same way as direct matching for $d^E$?) It would be better if the reason for this part was explicitly explained in the paper.
- How was the value of hyperparameter $\alpha$ selected? How sensitive is IOSTOM's performance to hyperparameter $\alpha$?
- The optimal value of the hyperparameter alpha seems to be different for each environment. Is there no way to automatically determine $\alpha$ (also for $\beta$)?
- LobsDICE is also a representative comparison algorithm that can be considered in the same setting. Why didn't you compare it? Can you add a comparison experiment?
- In the paper, it was explained that the absence of discriminator learning is an advantage, but instead, $V$ learning was introduced. Isn't it similar? What makes it better?
- Is it true that all the experimental learning was done with only offline datasets? Is it true that hyperparameter search and final model selection were also done without testing the environment?
- In all experiments, were you only training using the offline dataset? Was there no access to the test environment even during hyperparameter search and final model selection?

**Ethical Concerns:**

["NO or VERY MINOR ethics concerns only"]

**Final Justification:**

I have raised my rating accordingly (3: Borderline reject). I appreciate the author's detailed response, but concerns remain regarding the experimental aspects of the paper, making it still difficult to change the rating to accept.

**Limitations:**

Yes, the authors adequately addressed the limitations and potential negative societal impact of their work.

**Paper Formatting Concerns:**

There are no paper formatting concerns for this submission.

**Quality:**

3

**Strengths And Weaknesses:**

**Strengths**

- The proposed method is theoretically well-grounded, with clearly derived formulations based on flow constraints, Lagrangian duality, and Q-weighted behavior cloning.
- The method is evaluated across a wide range of environments and consistently achieves strong performance.

**Weaknesses**

- There is no experimental comparison with LobsDICE, one of the major baseline algorithms, considering the same problem setting.
- Since hyperparameter selection for the real environment is not possible in offline settings, analysis of hyperparameter alpha is thought to be necessary.
- It requires an additional hyperparameter beta, and its performance is very sensitive to $\beta$.
- It seems necessary to check whether fair experiments and comparisons were conducted in an offline setting. (ex. when determining hyperparameter $\alpha$ and $\beta$, whether the settings with the best performance in the test environment were selected and reported as the final results.)

---

> ### Author Rebuttal · Authors · 2025-07-30
>
> We thank reviewer for your helpful comments and feedback. We provide our responses below to address your concerns.
>
> > 1. What is the main reason for considering mixed state visitation distribution instead of starting from directly matching the objective with $d^E$? (What is the main problem if we derive it in the same way as direct matching for $d^E$?)...
>
> We thank the reviewer for the insightful question. The objective of directly matching between $d^{\pi}$ and $d^{E}$ requires on-policy samples from $\pi$ to solve which is not ideal in offline setting because these samples can be out-of-distribution. This issue is already discussed in previous LfO works like LobsDICE (Section 2.2 in [1]) and SMODICE (Section 3.1 in [2]). To mitigate this, these methods need to integrate the action-labeled suboptimal dataset into learning process by solving surgroate objectives as we note in **Section 3** of our paper.  Through our formulation in Eq. 2, we are also able to leverage the suboptimal dataset in training process. However, the key distinction is that our objective is mathematically constructed to have the same optimal solution as the original, direct-matching problem.
>
>
> > 2. How was the value of hyperparameter $\alpha$ selected? How sensitive is IOSTOM's performance to hyperparameter $\alpha$?
>
> **Please refer to Point 5 of the responses to reviewer 4HqZ**
>
> > 3. The optimal value of the hyperparameter alpha seems to be different for each environment. Is there no way to automatically determine $\alpha$(also for $\beta$)?
>
> We thank the reviewer for the questions. As pointed out in our response to Reviewer 4HqZ, $\alpha$ is not a sensitive parameter and can be fixed to 0.5 following previous works. The hyperparameter $\beta$, which controls the regularization strength in the extreme value learning framework, is often sensitive and may need to be tuned per environment to achieve the best performance, as shown in [3]. However, we found very minimal hyperparameter tuning was required in our framework. As shown in Appendix C.3, we fix $\beta$ to 20 for all locomotion tasks and $\beta$ to 10 for most manipulation tasks except for "pen-cloned" where β is set to 3. The ablation study of $\beta$ in Appendix D.3 also shows that in IOSTOM, $\beta$ is only sensitive per class of environment, not per individual environment. Specifically, high values like 15 or 20 are consistently effective for locomotion tasks, while lower values like 5 or 10 achieve good results on manipulation tasks. This provides a straightforward heuristic for setting this parameter: one can either select a value based on the task type (locomotion vs. manipulation) or perform a quick search on a single representative task from a class and apply that value to all others in the same class.
>
> > 4. LobsDICE is also a representative comparison algorithm that can be considered in the same setting. Why didn't you compare it?...
>
> We thank the reviewer for bringing up LobsDICE. Initially, we did not include it in our baselines because PW-DICE—a more recent and related method—was already shown in [4] to outperform LobsDICE on the “random+expert” D4RL setting.
>
> To address your question and provide a more complete comparison, we have conducted additional experiments comparing LobsDICE, PW-DICE, and our method (IOSTOM), as shown in the table below.
>
> The results clearly demonstrate that IOSTOM significantly outperforms LobsDICE across all tasks. Like other DICE-based methods, LobsDICE performs poorly on the “few-expert” and manipulation tasks, while our method remains robust. Our findings also reaffirm that PW-DICE surpasses LobsDICE in the “random+expert” scenario. However, PW-DICE underperforms in several other settings, highlighting the general instability of discriminator-based methods. These comparisons further emphasize the advantages of our stable, discriminator-free approach. We will include the full results in the updated paper.
>
> | Suboptimal Dataset | Env | PW-DICE | LobsDICE | IOSTOM |
> |---|---|---|---|---|
> | random+expert | hopper | 108.09±2.39 | 99.64+-5.01 | **109.32±1.08** |
> |  | halfcheetah | 86.11±4.39 | 80.76+-3.17 | **93.02±0.40** |
> |  | walker2d | 107.48±0.53 | 107.54+-0.13 | **107.98±0.20** |
> |  | ant | 126.89±1.17 | 122.65+-0.72 | **128.19±1.52** |
> | random+few-expert | hopper | 75.04±14.21 | 74.25+-11.23 | **107.28±3.92** |
> |  | halfcheetah | 4.02 ±1.74 | 11.61+-5.66 | **88.77±1.26** |
> |  | walker2d | 36.11 ±9.19 | 101.29+-5.53 | **108.40±0.21** |
> |  | ant | 99.90 ±2.59 | 93.84+-7.51 | **120.09±5.17** |
> | medium+expert | hopper | 65.99 ±8.05 | 74.43+-5.28 | **110.20±0.51** |
> |  | halfcheetah | 58.74 ±1.84 | 71.09+-3.76 | **93.12±0.32** |
> |  | walker2d | 105.41 ±0.33 | 103.34+-2.11 | **108.12±0.13** |
> |  | ant | 108.14±1.90 | 114.96+-4.05 | **124.72±3.49** |
> | medium+few-expert | hopper | 57.24±3.03 | 67.00+-6.43 | **108.96±1.33** |
> |  | halfcheetah | 27.85±6.03 | 44.76+-3.94 | **89.47±0.82** |
> |  | walker2d | 75.22±7.05 | 95.39+-5.22 | **108.15±0.43** |
> |  | ant | 90.34±2.56 | 95.33+-1.08 | **120.36±1.25** |
> | cloned+expert | pen | 23.39±4.56 | 29.83 +- 4.59 | **82.77±4.84** |
> |  | door | 0.07±0.14 | 0.02 +- 0.00 | **102.77±0.96** |
> |  | hammer | 1.29±0.12 | 0.55 +- 0.19 | **94.59±9.39** |
> | human+expert | pen | -2.56±1.30 | 42.09 +- 5.08 | **95.77±8.91** |
> |  | door | 0.15±0.02 | 10.98 +- 9.71 | **100.77±1.68** |
> |  | hammer | 2.02±0.77 | 17.06 +- 13.44 | **93.34±7.41** |
> | partial+expert | kitchen | 12.33±5.38 | 40.33 +- 2.24 | **58.95±2.27** |
> | mixed+expert | kitchen | 7.50±4.16 | 45.67 +- 2.75 | **46.45±0.84** |
>
>
> > 5. In the paper, it was explained that the absence of discriminator learning is an advantage, but instead $V$, learning was introduced. Isn't it similar? What makes it better?
>
> We thank the reviewer for the insightful question. In our setting, discriminator learning and value function learning serve fundamentally different purposes. The discriminator acts as a reward function, assigning high scores to expert states and low scores to non-expert ones. In contrast, our value function does more than binary classification—it imposes a structured ranking over suboptimal states by assigning higher values to those from which expert states are more easily reachable. This provides a meaningful recovery signal, guiding the agent back toward expert-like behavior when it deviates. Such nuanced guidance is something a simple reward-like discriminator cannot provide.
>
> > 6. In all experiments, were you only training using the offline dataset? Was there no access to the test environment even during hyperparameter search and final model selection?
>
> We thank the reviewer for the question. In our work, the entire training process for IOSTOM is conducted strictly offline, using only the provided datasets. This stands in contrast to online IL methods like GAIL [6], which require continuous interaction with the environment to populate a replay buffer.
>
> However, we do require limited access to the environment for hyperparameter tuning and performance evaluation. This is a standard practice in both offline RL (e.g., Table 4 in [3]) and offline IL (e.g., Table 7 in [5]), where returns from the environment are used to select the best-performing hyperparameters for each task or task class.
>
> While fully offline alternatives exist, they are less applicable in our setting. For example, off-policy evaluation (OPE) methods [7] are commonly used in offline RL but require reward information, which is not available in our imitation learning setup. Another alternative—using validation loss to guide selection—is often unreliable due to its poor correlation with final policy performance [8].
>
> Therefore, we adopt the commonly accepted assumption of having limited evaluation access to the environment. In our case, since $\alpha$ is fixed at 0.5, we only tune $\beta$ from a small candidate set {3, 5, 7, 15, 20}, requiring at most five evaluation trials.
>
> In terms of your concern about fair comparisions where we might change the $\alpha$ and $\beta$ per environment to beat other approaches, we want to note that we keep the same $\beta=20$ across all locomotion tasks. We only use different values of $\beta$ ($\beta=3$ for "pen-cloned" task and $\beta=10$ for the rest) when solving manipulation tasks. When dealing with different type of tasks like manipulation, previous LfO methods like SMODICE or PW-DICE also needs to modify their agorithmic choices: changing from KL divergence to $\chi^2$ divergence in SMODICE (Appendix E.2 in [2]) and change in hyperparameter in PW-DICE (Table 1 in [4]). If we set a consistent $\beta=10$ for all manipulation tasks, IOSTOM is still the leading approach 7/8 tasks. Therefore, we believe that IOSTOM outperforms all baselines under a fair and standard comparison protocol.
>
> **References:**
>
> [1]: Kim, Geon-Hyeong, et al. "Lobsdice: Offline learning from observation via stationary distribution correction estimation." NeurIPS 2022.
>
> [2]: Ma, Yecheng, et al. "Versatile offline imitation from observations and examples via regularized state-occupancy matching." ICML 2022.
>
> [3]: Garg, Divyansh, et al. "Extreme q-learning: Maxent rl without entropy."  ICLR 2023.
>
> [4]: Yan, Kai, Alexander G. Schwing, and Yu-Xiong Wang. "Offline imitation from observation via primal wasserstein state occupancy matching." ICML 2024.
>
> [5]: Chi, Cheng, et al. "Diffusion policy: Visuomotor policy learning via action diffusion." The International Journal of Robotics Research (2023)
>
> [6]: Ho et al. "Generative adversarial imitation learning." NeurIPS 2016.
>
> [7]: Fu, Justin, et al. "Benchmarks for deep off-policy evaluation." ICLR 2021.
>
> [8]: Mandlekar, Ajay, et al. "What matters in learning from offline human demonstrations for robot manipulation." CoRL 2021.
>
> ---
>
> **We hope our revisions and our response  address the reviewers’ concerns and further clarify our contributions. If there are any additional questions or comments, we would be happy to address them.**

---

> > ### Author Response · Authors · 2025-08-06
> >
> > Dear Reviewer y8cD,
> >
> > Thank you again for your constructive feedback. With the author-reviewer discussion deadline approaching in about 75 hours, we kindly invite you to consider our rebuttal. We believe we have addressed all your concerns and hope our revisions are satisfactory.
> >
> > Thank you for your time and consideration.

---

> > > ### Comment · Area_Chair_hnFe · 2025-08-06
> > >
> > > Dear reviewer y8cD,
> > >
> > > Could you take time to respond to authors' rebuttal as soon as possible?
> > >
> > > Thank you!
> > >
> > > AC

---

> ### Comment · Reviewer_y8cD · 2025-08-06
> **Official Comment by Reviewer y8cD**
>
> Thank you for the detailed response from the authors. Most of my questions were answered through the author's response. However, I do not completely agree with the claim that it is not sensitive to the hyperparameter alpha, and I still have concerns that there is no guarantee that it will work regardless of the value of alpha when experimenting in a new domain. I will also check all the other reviews and responses and decide whether to change the ratings or not.

---

> > ### Author Response · Authors · 2025-08-07
> >
> > We thank the reviewer for reading our response and providing additional valuable feedback.
> >
> > While we agree that there is no guarantee that the selected hyperparameter value for $\alpha$ will generalize to entirely new domains (which is a common practice in most ML research), we believe our extensive ablation studies (span a wide range of tasks across nearly all commonly used benchmarks in imitation learning) offer meaningful insight into its effect. We will definitely incorporate these additional experiments into the final version of the paper.
> >
> > We hope this helps clarify our position. Once again, we sincerely appreciate your time and thoughtful evaluation. We kindly hope that you will take these points into consideration during your final assessment.

---

### Official Review · Reviewer_4HqZ · 2025-06-28

**Clarity:** 3
**Significance:** 3
**Originality:** 3
**Rating:** 4
**Confidence:** 4

**Summary:**

The paper proposes IOSTOM, an offline imitation learning from observation method based on state transition occupancy matching that is discriminator-free and data-efficient. The proposed method operates on a state-only expert dataset and state-action suboptimal dataset. The proposed method consists of several steps. First, IOSTOM recovers a state-transition probability distribution ("implicit policy") by minimizing $\chi^2$-divergence between the learned state-transition and expert state-transition (regularized by suboptimal dataset policy), which can be converted to an unconstrained bilevel Q-value optimization via Lagrange multipliers and convex conjugates. Soft value estimates are adopted to avoid Out-Of-Distribution (OOD) samples. Finally, the policy is retrieved by advantage-weighted behavior cloning from the suboptimal state-action dataset (as a lower bound surrogate for optimizing implicit policy with Inverse-Dynamic Model (IDM)). The proposed algorithm is evaluated on multiple environments which outperforms prior methods.

**Questions:**

I have several questions:

1. Is it possible to add a teaser figure for this paper? I feel people who are less familiar with the state occupancy matching framework can understand the idea more easily with an illustration.

2. Is there a more direct ablation / empirical evidence (e.g. visualization) to show in some cases discriminators fail other than final performance difference?

3. Why do the authors choose $\alpha=0.5$ for distribution mixture with suboptimal dataset, which seems quite large? Can the authors provide an ablation on different values of $\alpha$?

4. Why do the authors choose to update policy during the training loop of other components (as shown in Alg. 1) instead of finishing the other parts of training first, which could potentially cause instability? $\theta$ is not involved in other parts of training after all.

**Ethical Concerns:**

["NO or VERY MINOR ethics concerns only"]

**Final Justification:**

The authors' response has addressed my concern satisfactorily. However, as some other reviewers feel their concern is unaddressed, I will keep my current score.

**Limitations:**

Yes.

**Quality:**

3

**Strengths And Weaknesses:**

**Strengths**

1. The paper is well-written and easy to follow; though the paper is math-heavy, the high-level idea behind the formulae (first optimize state-pair divergence, then use Lagrange multiplier & convex conjugate to convert to unconstrained problems, then soft values to avoid OOD issues, and finally extract policy via lower bound of implicit policy with IDM) is very clearly conveyed.

2. The idea of this paper is interesting and well-supported by mathematical derivations: bypassing discriminators by optimizing "implicit policy", which is a novel notion.

3. The evaluation results look solid with sufficient amount of ablations. I particularly appreciate the video demonstrations in the supplementary material.

**Weaknesses**

1. the training objective in Eq. 9 seems potentially unstable, as there is a term $(Q(s,s')-\frac{2}{1-\gamma})^2$. When $\gamma$ is close to 1 with long horizon tasks, this can lead to a very large value of $Q$ which causes numerical issues during optimization.

2. The number of expert dataset used in the experiment is quite large, especially for the MuJoCo datasets. In the appendix, the authors stated that they use 30 expert trajectories for "few-expert" scenario. In contrast, both SMODICE (https://arxiv.org/pdf/2202.02433, Sec. 6.1) and PW-DICE (https://arxiv.org/pdf/2311.01331, Appendix D.2) uses **single** expert trajectory. I would like to see how the proposed algorithm work with even less amount of expert trajectories than the current setting.

**Minor Weaknesses**

1. Some reference format are incorrect. For example: "[34] introduce ReCOIL" -> "Sikchi et al. [34] introduce ReCOIL"; "Following [40]" -> "Following Xu et al. [40]".

2. The period in Tab. 4, Tab. 5 and Fig. 3 caption is missing.

3. The main paper uses "inverse dynamic model", but the appendix line 648 uses "inverse kinematics model".

---

> ### Author Rebuttal · Authors · 2025-07-30
>
> We thank the reviewer for the constructive comments and valuable feedback. Below, we provide our detailed responses to address your concerns.
>
> > 1. The training objective in Eq. 9 seems potentially unstable, as there is a term $(Q(s, s')-\frac{2}{1-\gamma})^2$. When $\gamma$ is close to 1 with long horizon tasks, this can lead to a very large value of $Q$ which causes numerical issues during optimization.
>
> We thank the reviewer for the insightful comment. In our work, the discount factor $\gamma$ is set to 0.99 (see Table 5 in the Appendix), which is a standard choice in RL. With $\gamma = 0.99$, the term $(Q(s, s') - \frac{2}{1 - \gamma})^2$ becomes $(Q(s, s') - 200)^2$. Our training remains stable with this formulation. Prior imitation learning methods such as LS-IQ [1] have also adopted this formulation to promote stable learning. The goal of  $(Q(s, s') - \frac{2}{1 - \gamma})^2$ is only assigning large and bounded $Q$ values to expert transitions as pointed in Section 5 and Appendix B.8 of our work. Therefore,  one could simply cap the target value (e.g. at 200) in the case of $\gamma$ > 0.99 (which is uncommon in deep RL). This would preserve the objective's intent to assign a high and stable Q-value to expert data while avoiding any potential numerical issues.
>
>
> > 2. The number of expert dataset used in the experiment is quite large, especially for the MuJoCo datasets. In the appendix, the authors stated that they use 30 expert trajectories for "few-expert" scenario. In contrast, both SMODICE (https://arxiv.org/pdf/2202.02433, Sec. 6.1) and PW-DICE (https://arxiv.org/pdf/2311.01331, Appendix D.2) uses single expert trajectory. I would like to see how the proposed algorithm work with even less amount of expert trajectories than the current setting.
>
> We thank the reviewer for raising this point. We believe there may be a misunderstanding. In Appendix C.1, we stated: *“The state-only expert dataset in all tasks includes only one expert trajectory.”* In our paper, the term “few-expert” refers to a suboptimal dataset constructed by combining "random" or "medium" trajectories with 30 expert trajectories.
>  This is similar to the experimental setting used in SMODICE and PW-DICE, where the suboptimal dataset contains 200 expert trajectories alongside lower-quality ones. The key difference across scenarios lies in how the suboptimal dataset is constructed, as detailed in **Table 4** of the Appendix.
>
> We would greatly appreciate it if the reviewer could point us to the specific part that caused confusion, so we can revise it for greater clarity.
>
> > 3. Is it possible to add a teaser figure for this paper? I feel people who are less familiar with the state occupancy matching framework can understand the idea more easily with an illustration.
>
> We thank the reviewer for the valuable suggestion, which could indeed improve the readability of the paper. We will include such a teaser figure in the revised version. Due to the NeurIPS rebuttal policy, we are unable to upload images at this stage, but it will be added in the updated version of the paper.
>
> > 4. Is there a more direct ablation / empirical evidence (e.g. visualization) to show in some cases discriminators fail other than final performance difference?
>
> We thank the reviewer for the valuable suggestion. Visualizing the reward landscape is indeed a helpful way to illustrate how the discriminator assigns high rewards to non-expert states in tasks where discriminator-based methods perform poorly. As the NeurIPS review system does not allow image uploads during the rebuttal phase, we will include these visualizations in the revised version of the paper.
>
>
> > 5. Why do the authors choose $\alpha =0.5$ for distribution mixture with suboptimal dataset, which seems quite large? Can the authors provide an ablation on different values of $\alpha$?
>
> We thank the reviewer for the insightful question. We set $\alpha = 0.5$ as it is a commonly adopted choice in prior works [1, 2, 3] that employ mixture distributions for $f$-divergence minimization.
>
> In response to your question, we conducted an ablation study on the sensitivity of $\alpha$, testing five values across all 24 tasks. The table below reports performance for each setting, with the best-performing $\alpha$ for each task highlighted in bold. We also include two additional metrics: $gap_{worst}$, representing the percentage gap between the best and worst $\alpha$, and $gap_{default}$, indicating the gap between the best-performing $\alpha$ and our default setting of $\alpha = 0.5$.
>
> According to the table, the average $gap_{worst}$ value across all tasks is just 4.54%, which is relatively small. This indicates that IOSTOM's performance is not highly sensitive to the choice of $\alpha$ . Furthermore, the average performance gap between the task-specific optimal $\alpha$ and our default choice of $\alpha=0.5$ is even smaller at 2.07%. The $gap_{default}$ value is also under 5% in all but 3 of 24 tasks. This confirms that $\alpha=0.5$ is a robust and effective hyperparameter choice, consistently providing near-optimal performance.
>
> | Suboptimal Dataset | Env | $\alpha=0.1$ | $\alpha=0.3$ | $\alpha=0.5$ (our) | $\alpha=0.7$ | $\alpha=0.9$ | $gap_{worst}$(%) | $gap_{default}$(%) |
> |---|---|---|---|---|---|---|---|---|
> | random+expert | hopper | 109.72+-0.84 | 110.21+-0.68 | 109.32+-1.08 | 109.19+-0.21 | **110.29+-0.64** | 1.00 | 0.88 |
> |  | halfcheetah | **93.09+-0.22** | 92.90+-0.31 | 93.02+-0.40 | 92.99+-0.08 | 92.89+-0.18 | 0.21 | 0.08 |
> |  | walker2d | 107.82+-0.23 | 107.97+-0.08 | 107.98+-0.20 | 108.16+-0.17 | **108.38+-0.26** | 0.52 | 0.37 |
> |  | ant | **128.37+-1.89** | 126.67+-2.89 | 128.19+-1.52 | 125.78+-3.00 | 127.30+-1.85 | 2.02 | 0.14 |
> | random+few-expert | hopper | 107.41+-1.27 | **108.59+-2.04** | 107.28+-3.92 | 104.75+-3.36 | 107.25+-4.11 | 3.54 | 1.21 |
> |  | halfcheetah | 87.29+-0.73 | 87.52+-2.11 | **88.77+-1.26** | 88.42+-1.00 | 86.61+-1.84 | 2.43 | 0.00 |
> |  | walker2d | 108.09+-0.24 | 108.21+-0.06 | 108.40+-0.21 | **108.45+-0.12** | 108.24+-0.14 | 0.33 | 0.05 |
> |  | ant | **125.43+-2.77** | 121.19+-2.09 | 120.09+-5.17 | 122.18+-1.90 | 123.46+-1.17 | 4.26 | 4.26 |
> | medium+expert | hopper | 109.79+-0.95 | 110.44+-0.51 | 110.20+-0.51 | 110.30+-0.54 | **110.61+-0.07** | 0.74 | 0.37 |
> |  | halfcheetah | **93.16+-0.19** | 92.82+-0.32 | 93.12+-0.32 | 93.00+-0.23 | 92.68+-0.17 | 0.52 | 0.04 |
> |  | walker2d | 107.54+-0.45 | 107.96+-0.12 | 108.12+-0.13 | 107.28+-1.44 | **108.22+-0.27** | 0.87 | 0.09 |
> |  | ant | **129.00+-0.59** | 124.91+-2.56 | 124.72+-3.49 | 125.52+-2.26 | 127.86+-2.20 | 3.32 | 3.32 |
> | medium+few-expert | hopper | 107.95+-1.72 | **110.08+-1.12** | 108.96+-1.33 | 106.99+-4.23 | 104.34+-5.88 | 5.21 | 1.02 |
> |  | halfcheetah | 87.98+-1.17 | 87.79+-1.77 | **89.47+-0.82** | 88.66+-1.31 | 89.17+-0.60 | 1.88 | 0.00 |
> |  | walker2d | 108.15+-0.31 | 108.24+-0.33 | 108.15+-0.43 | 108.16+-0.22 | **108.47+-0.22** | 0.30 | 0.30 |
> |  | ant | 120.44+-1.57 | 119.36+-1.84 | 120.36+-1.25 | 119.12+-0.58 | **122.32+-0.73** | 2.62 | 1.60 |
> | cloned+expert | pen | 50.08+-17.30 | 72.62+-7.45 | **82.77+-4.84** | 73.96+-5.58 | 76.37+-6.52 | 39.49 | 0.00 |
> |  | door | 102.68+-0.63 | **103.79+-0.63** | 102.77+-0.96 | 102.65+-1.37 | 103.64+-0.92 | 1.10 | 0.98 |
> |  | hammer | 101.98+-4.88 | **110.07+-7.29** | 94.59+-9.39 | 105.58+-12.35 | 106.81+-2.94 | 14.06 | 14.06 |
> | human+expert | pen | 95.75+-7.18 | 96.68+-4.83 | 95.77+-8.91 | **98.29+-5.11** | 97.74+-5.11 | 2.58 | 2.56 |
> |  | door | 100.41+-3.85 | 100.94+-3.65 | 100.77+-1.68 | 99.99+-2.31 | **101.13+-1.07** | 1.13 | 0.36 |
> |  | hammer | 95.24+-5.79 | 101.71+-4.97 | 93.34+-7.41 | 101.04+-6.36 | **103.79+-8.80** | 10.07 | 10.07 |
> | partial+expert | kitchen | **64.00+-7.66** | 61.83+-1.38 | 58.95+-2.27 | 61.33+-4.13 | 60.08+-3.21 | 7.89 | 7.89 |
> | mixed+expert | kitchen | 45.25+-2.38 | 45.42+-4.05 | **46.45+-0.84** | 45.08+-1.13 | 45.33+-2.01 | 2.95 | 0.00 |
> | Average |  |  |  |  |  |  | 4.54 | 2.07 |
>
> > 6. Why do the authors choose to update policy during the training loop of other components (as shown in Alg. 1) instead of finishing the other parts of training first, which could potentially cause instability? $\theta$ is not involved in other parts of training after all.
>
> We thank the reviewer for the insightful question. The reason is that the policy does not influence the value function updates, so policy extraction can be performed either concurrently with or after value learning. In our implementation, we follow the XQL [4] codebase, which updates the policy jointly with the value networks. This joint training approach is also adopted by other LfO methods such as DILO and LobsDICE (see Section 4.3 of [3] and Appendix G.2 of [5]).
>
> **References:**
>
> [1]: Al-Hafez, Firas, et al. "Ls-iq: Implicit reward regularization for inverse reinforcement learning." ICLR 2023.
>
> [2]: Sikchi, Harshit, et al. "Dual rl: Unification and new methods for reinforcement and imitation learning." ICLR 2024.
>
> [3]: Sikchi, Harshit, et al. "A dual approach to imitation learning from observations with offline datasets." CoRL 2024.
>
> [4]: Garg, Divyansh, et al. "Extreme q-learning: Maxent rl without entropy."  ICLR 2023.
>
> [5]: Kim, Geon-Hyeong, et al. "Lobsdice: Offline learning from observation via stationary distribution correction estimation." NeurIPS 2022.
>
>
> ---
>
> **We hope our revisions and our response  address the reviewers’ concerns and further clarify our contributions. If there are any additional questions or comments, we would be happy to address them.**

---

> > ### Comment · Reviewer_4HqZ · 2025-08-01
> >
> > Thanks for the authors' detailed response. I feel the rebuttal addressed my concerns and I do not have other questions. I will wait and check if other reviewers feel their concern being addressed and decide whether to keep the current score or increase my score.

---

> > > ### Author Response · Authors · 2025-08-08
> > > **Thank you for the feedback!**
> > >
> > > We sincerely thank the reviewer for carefully reading our paper, providing thoughtful responses, and offering valuable feedback. We will certainly incorporate the corresponding discussions and the additional experiments to the final version of the paper.
> > >
> > > We remain fully available and would be glad to provide any further details or clarify any remaining points. We believe our responses have addressed most of the concerns raised by you and the other reviewers.
> > >
> > > If our replies and the planned revisions have satisfactorily resolved your concerns, we would be grateful if you could kindly reflect this in your evaluation of our work.
> > >
> > > Thank you once again for your constructive engagement!

---

### Official Review · Reviewer_gGMk · 2025-06-29

**Clarity:** 3
**Significance:** 3
**Originality:** 3
**Rating:** 5
**Confidence:** 2

**Summary:**

This paper introduces a novel approach (IOSTOM) for offline LfO problem by joint state visitation distribution matching. This method specifically avoids using discriminators or inverse dynamics models, and instead learns an implicit policy via state-to-state transition probabilities without using action label. Empirical results on D4RL datasets demonstrate the efficacy of the method.

**Questions:**

What is the intuition of using a mixture of joint state visitation distributions $d_\mathrm{mix}^I, d_\mathrm{mix}^{E,I}$? How does the mixing parameter $\alpha$ influences the training? It would be helpful to introduce the intuition before deriving the full equation.

**Ethical Concerns:**

["NO or VERY MINOR ethics concerns only"]

**Final Justification:**

Thank you for the detailed replies and additional experiments. I have no further concerns and am keeping my score.

**Limitations:**

Yes

**Quality:**

3

**Strengths And Weaknesses:**

**Strengths:**

- The paper presents a novel LfO approach grounded in occupancy matching.

- The theoretical framework and derivation of the algorithm is very solid.

- The method is proved to be  practical and efficient with its discriminator-free and IDM-free design.

- Demonstrates better performance on diverse and challenging benchmarks, outperforming all baselines, especially inclusing an interesting real-world case study on marine navigation.

- Demonstrates robustness to low-data settings and sub-sampling.



**Weaknesses:**

* Access to fully observable actions in sub-optimal dataset is hard to guarantee. It would be helpful to discuss how realistic it is.

---

> ### Author Rebuttal · Authors · 2025-07-30
>
> We thank reviewer for your helpful comments and feedback. We provide our responses below to address your concerns.
>
> > 1. Access to fully observable actions in sub-optimal dataset is hard to guarantee. It would be helpful to discuss how realistic it is.
>
> We thank the reviewer for the insightful comments. We believe that access to action-labeled suboptimal datasets is highly realistic in many practical settings. For example, in the robotics community, collecting demonstrations via teleoperation from non-expert users is often significantly easier and more cost-effective than acquiring expert demonstrations, which typically require domain-specific training or expert operators [1].
>
> Beyond robotics, similar trends are observed in other domains. In autonomous driving, for instance, large-scale datasets are often gathered from regular human drivers who may not always exhibit optimal behavior. These action-labeled suboptimal logs still provide valuable data for training and evaluation. In healthcare applications, clinician interactions recorded in electronic health records can be viewed as suboptimal demonstrations, yet they form the basis for many decision-support models. Similarly, in gaming and digital environments, user gameplay logs—despite being noisy or suboptimal—are widely used for imitation and reinforcement learning research.
>
> These examples underscore that suboptimal, action-labeled data are not only easier to obtain but also more abundant and representative of real-world behavior. As such, they offer a practical and scalable foundation for learning from demonstrations in many domains.
>
> We will incorporate these examples into the revised paper to better illustrate the practicality of accessing action-labeled suboptimal datasets.
>
> > 2. What is the intuition of using a mixture of joint state-visitation distributions $d_{mix}^I, d_{mix}^{E,I}$? It would be helpful to introduce the intuition before deriving the full equation.
>
> We thank the reviewer for the insightful question. The use of a mixture of joint state-visitation distributions is adopted from some recent state-of-the-art papers on RL and IL (e.g., DualRL [2]), which highlights its advantage in avoiding on-policy sampling.
>
> Specifically, prior work typically adopts matching between $d^{\pi}$ and $d^{E}$, but this approach requires on-policy samples from $\pi$ to solve. This is not ideal in the offline setting, as these samples can be out-of-distribution. This limitation has been discussed in previous LfO works such as LobsDICE (Section 2.2 in [3]) and SMODICE (Section 3.1 in [4]). Using a mixture of joint state-visitation distributions helps address this issue by avoiding reliance on on-policy data while maintaining effective alignment between expert and policy behavior. We will include this discussion earlier in the updated paper to better clarify the motivation behind this design choice.
>
>
> > 3. How does the mixing parameter $\alpha$ influence the training?
>
> We thank the reviewer for the question. In response, we conducted an ablation study on the sensitivity of $\alpha$, testing five values across all 24 tasks. The table below reports performance for each setting, with the best-performing $\alpha$ for each task highlighted in bold. We also include two additional metrics: $gap_{worst}$, representing the percentage gap between the best and worst $\alpha$, and $gap_{default}$, indicating the gap between the best-performing $\alpha$ and our default setting of $\alpha = 0.5$.
>
> According to the table, the average $gap_{worst}$ value across all tasks is just 4.54%, which is relatively small. This indicates that IOSTOM's performance is not highly sensitive to the choice of $\alpha$. Furthermore, the average performance gap between the task-specific optimal $\alpha$ and our default choice of $\alpha=0.5$ is even smaller at 2.07%. The $gap_{default}$ value is also under 5% in all but 3 of 24 tasks. This confirms that $\alpha=0.5$ is a robust and effective hyperparameter choice, consistently providing near-optimal performance.
>
> | Suboptimal Dataset | Env         | $\alpha=0.1$     | $\alpha=0.3$     | $\alpha=0.5$ (our)    | $\alpha=0.7$     | $\alpha=0.9$     | $gap_{worst}$(%) | $gap_{default}$(%) |
> |--------------------|-------------|------------------|------------------|-----------------|------------------|------------------|---------------|-----------------|
> | random+expert      | hopper      | 109.72+-0.84     | 110.21+-0.68     | 109.32+-1.08    | 109.19+-0.21     | **110.29+-0.64** | 1.00          | 0.88            |
> |                    | halfcheetah | **93.09+-0.22**  | 92.90+-0.31      | 93.02+-0.40     | 92.99+-0.08      | 92.89+-0.18      | 0.21          | 0.08            |
> |                    | walker2d    | 107.82+-0.23     | 107.97+-0.08     | 107.98+-0.20    | 108.16+-0.17     | **108.38+-0.26** | 0.52          | 0.37            |
> |                    | ant         | **128.37+-1.89** | 126.67+-2.89     | 128.19+-1.52    | 125.78+-3.00     | 127.30+-1.85     | 2.02          | 0.14            |
> | random+few-expert  | hopper      | 107.41+-1.27     | **108.59+-2.04** | 107.28+-3.92    | 104.75+-3.36     | 107.25+-4.11     | 3.54          | 1.21            |
> |                    | halfcheetah | 87.29+-0.73      | 87.52+-2.11      | **88.77+-1.26** | 88.42+-1.00      | 86.61+-1.84      | 2.43          | 0.00            |
> |                    | walker2d    | 108.09+-0.24     | 108.21+-0.06     | 108.40+-0.21    | **108.45+-0.12** | 108.24+-0.14     | 0.33          | 0.05            |
> |                    | ant         | **125.43+-2.77** | 121.19+-2.09     | 120.09+-5.17    | 122.18+-1.90     | 123.46+-1.17     | 4.26          | 4.26            |
> | medium+expert      | hopper      | 109.79+-0.95     | 110.44+-0.51     | 110.20+-0.51    | 110.30+-0.54     | **110.61+-0.07** | 0.74          | 0.37            |
> |                    | halfcheetah | **93.16+-0.19**  | 92.82+-0.32      | 93.12+-0.32     | 93.00+-0.23      | 92.68+-0.17      | 0.52          | 0.04            |
> |                    | walker2d    | 107.54+-0.45     | 107.96+-0.12     | 108.12+-0.13    | 107.28+-1.44     | **108.22+-0.27** | 0.87          | 0.09            |
> |                    | ant         | **129.00+-0.59** | 124.91+-2.56     | 124.72+-3.49    | 125.52+-2.26     | 127.86+-2.20     | 3.32          | 3.32            |
> | medium+few-expert  | hopper      | 107.95+-1.72     | **110.08+-1.12** | 108.96+-1.33    | 106.99+-4.23     | 104.34+-5.88     | 5.21          | 1.02            |
> |                    | halfcheetah | 87.98+-1.17      | 87.79+-1.77      | **89.47+-0.82** | 88.66+-1.31      | 89.17+-0.60      | 1.88          | 0.00            |
> |                    | walker2d    | 108.15+-0.31     | 108.24+-0.33     | 108.15+-0.43    | 108.16+-0.22     | **108.47+-0.22** | 0.30          | 0.30            |
> |                    | ant         | 120.44+-1.57     | 119.36+-1.84     | 120.36+-1.25    | 119.12+-0.58     | **122.32+-0.73** | 2.62          | 1.60            |
> | cloned+expert      | pen         | 50.08+-17.30     | 72.62+-7.45      | **82.77+-4.84** | 73.96+-5.58      | 76.37+-6.52      | 39.49         | 0.00            |
> |                    | door        | 102.68+-0.63     | **103.79+-0.63** | 102.77+-0.96    | 102.65+-1.37     | 103.64+-0.92     | 1.10          | 0.98            |
> |                    | hammer      | 101.98+-4.88     | **110.07+-7.29** | 94.59+-9.39     | 105.58+-12.35    | 106.81+-2.94     | 14.06         | 14.06           |
> | human+expert       | pen         | 95.75+-7.18      | 96.68+-4.83      | 95.77+-8.91     | **98.29+-5.11**  | 97.74+-5.11      | 2.58          | 2.56            |
> |                    | door        | 100.41+-3.85     | 100.94+-3.65     | 100.77+-1.68    | 99.99+-2.31      | **101.13+-1.07** | 1.13          | 0.36            |
> |                    | hammer      | 95.24+-5.79      | 101.71+-4.97     | 93.34+-7.41     | 101.04+-6.36     | **103.79+-8.80** | 10.07         | 10.07           |
> | partial+expert     | kitchen     | **64.00+-7.66**  | 61.83+-1.38      | 58.95+-2.27     | 61.33+-4.13      | 60.08+-3.21      | 7.89          | 7.89            |
> | mixed+expert       | kitchen     | 45.25+-2.38      | 45.42+-4.05      | **46.45+-0.84** | 45.08+-1.13      | 45.33+-2.01      | 2.95          | 0.00            |
> | Average            |             |                  |                  |                 |                  |                  | 4.54          | 2.07            |
>
> **References:**
>
> [1]: Lynch, Corey, et al. "Learning latent plans from play." Conference on robot learning. Pmlr, 2020.
>
> [2]: Sikchi, Harshit, et al. "Dual rl: Unification and new methods for reinforcement and imitation learning." ICLR 2024.
>
> [3]: Kim, Geon-Hyeong, et al. "Lobsdice: Offline learning from observation via stationary distribution correction estimation." NeurIPS 2022.
>
> [4]: Ma, Yecheng, et al. "Versatile offline imitation from observations and examples via regularized state-occupancy matching." ICML 2022.
>
> ---
>
> **We hope our revisions and our response  address the reviewers’ concerns and further clarify our contributions. If there are any additional questions or comments, we would be happy to address them.**

---

> ### Author Response · Authors · 2025-08-08
>
> Dear Reviewer gGMk
>
> We sincerely thank the reviewer for carefully reading our paper, providing a thoughtful response, and offering valuable feedback. We will certainly incorporate the corresponding discussions and the additional experiments. We remain fully available and would be glad to provide any further details or clarify any remaining points.
>
> We believe our responses have addressed most of the concerns raised by you and the other reviewers. If our replies and the planned revisions have satisfactorily resolved your concerns, we would be grateful if you could kindly reflect this in your evaluation of our work.
>
> Thank you once again for your constructive engagement.

---

### Official Review · Reviewer_ahUM · 2025-07-01

**Clarity:** 1
**Significance:** 3
**Originality:** 3
**Rating:** 4
**Confidence:** 2

**Summary:**

This paper introduces ISOTOM, a framework for offline imitation learning from observation (IfO). ISOTOM aims to address the offline IfO problem while improving training stability by eliminating the need for a discriminator and an inverse dynamics model. Specifically, ISOTOM relies on state-transition distribution matching to remove dependence on action information. The paper first presents a joint-state Q-learning formulation to recover the implicit policy, followed by policy learning through advantage-weighted behavioral cloning. Empirical results demonstrate that ISOTOM outperforms baseline methods in MuJoCo environments and a marine navigation scenario.

**Questions:**

- Have the authors conducted any experiments with subsampled random data to assess the sensitivity of the method to the amount of random data?
- Have the authors conducted any experiments in image-based environments? Such settings may better reflect the nature of imitation from observation scenarios, where actions are not available and only visual state information is provided.

I am willing to increase the score if most of the concerns in the weakness and questions sections are solved.

**Ethical Concerns:**

["NO or VERY MINOR ethics concerns only"]

**Final Justification:**

As I mentioned in my comment, the authors have addressed the concerns raised in the review, particularly by conducting experiments to demonstrate the robustness of the proposed method. Additionally, the clarity issue does not appear to be a significant problem for other reviewers. Therefore, I am inclined to raise my overall score to 4 (borderline accept).

**Limitations:**

Yes (in the appendix)

**Paper Formatting Concerns:**

No paper formatting concerns found.

**Quality:**

3

**Strengths And Weaknesses:**

[Strengths]

(1) Quality
- The paper provides both theoretical analyses and empirical results demonstrating the effectiveness of the proposed method. However, I did not examine the proofs in the appendix in detail.
- The experiment section includes numerical results not only for MuJoCo simulated environments but also for a real-world marine navigation dataset.

(3) Significance
- The proposed method demonstrates superior performance compared to the baselines in both MuJoCo simulation environments and a real-world marine navigation scenario.

(4) Originality
- The paper introduces an objective function that excludes actions, which appears to be a novel contribution in the context of offline imitation from observation.

[Weaknesses]

(2) Clarity
- I believe the primary issue that makes me hesitant to recommend solid acceptance is the lack of clarity in the paper. While this may partly reflect my own limitations in expertise, the presentation feels unstructured and densely written, which hinders comprehension. For example:
    - In Section 3, the definitions of $d^E(s)$ and $d^I(s)$ are missing or unclear.
    - In Section 4.1, $d_0(s)$ is not properly defined.
    - The transition from Equation (3) to (4) is not clearly explained. I was able to derive (4) from (3), but it required effort that could have been avoided with a brief clarification.
    - The paper does not specify where the proofs of the main theorem can be found, even though they are in the appendix.
    - Additionally, several parts of the text (e.g., lines 121, 124, 146, 148, 150, 225, 235, 244) are written in a way that makes them difficult to parse.

These issues in notation, definition, and mathematical transitions collectively hindered my ability to follow the technical development of the paper. As a result, it was difficult to clearly identify the main contributions and understand the reasons behind the strengths of the proposed method compared to prior work. Improving the logical flow between sections and organizing the key concepts more clearly would significantly enhance readability and overall comprehension.

---

> ### Author Rebuttal · Authors · 2025-07-30
>
> We thank reviewer for your helpful comments and feedback. We provide our responses below to address your concerns.
>
> > 1. Issues about paper's clarity
>
> We thank the reviewer for the valuable comments regarding the clarity of our paper. We will revise the manuscript accordingly based on your suggestions.
> > In Section 3, the definitions of $d^E(s), d^I(s)$ and are missing or unclear.
>
> They are occupancy measures of the expert and sup-optimal datasets. We will clarify them.
>
> > In Section 4.1, $d_0(s)$ is not properly defined.
>
> This is the visitation distribution of  state $s$ at time 0. We wil clarify this.
>
> >The transition from Equation (3) to (4) is not clearly explained. I was able to derive (4) from (3), but it required effort that could have been avoided with a brief clarification.
>
> We have removed some intermediate derivations due to space limitations. We will clarify this transition in the revised version.
>
> > The paper does not specify where the proofs of the main theorem can be found, even though they are in the appendix.
>
> Thank you for the comment. We will add references to the corresponding proofs in the appendix.
> > Additionally, several parts of the text (e.g., lines 121, 124, 146, 148, 150, 225, 235, 244) are written in a way that makes them difficult to parse.
>
> Thank you for pointing out these detailed issues. These parts were compressed due to space limitations, which may have caused some lack of clarity. We will revise them to make the content clearer and easier to follow.
>
> > 2. Have the authors conducted any experiments with subsampled random data to assess the sensitivity of the method to the amount of random data?
>
> We thank the reviewer for the insightful question. In response, we applied subsampling rates of 5 and 20, meaning only 20% and 5% of the low-quality trajectories ("random" or "medium") from the original suboptimal dataset were used, respectively. The table below compares the performance of IOSTOM(full) with these subsampled variants—IOSTOM(sub5) and IOSTOM(sub20). Drops in performance exceeding 5% relative to the full dataset are highlighted in bold.
>
>
> When reducing the low-quality data to 20% of its original size (IOSTOM(sub5)), our method demonstrates strong robustness. Across all tasks and datasets, the performance remains very close to the results on the full dataset across all tasks with no significant degradation observed. When the low-quality data is aggressively subsampled to just 5% (IOSTOM(sub20)), we observe a more noticeable performance drop (up to about 18%), particularly in th halfcheetah and ant environments. However, IOSTOM is still robust on 10/16 tasks in this extremely limited data scenario.
>
> | Suboptimal Dataset | Env         | IOSTOM(full) | IOSTOM(sub5) | IOSTOM(sub20)    |
> |--------------------|-------------|--------------|--------------|------------------|
> | random+expert      | hopper      | 109.32±1.08  | 110.24+-0.63 | 110.66+-0.16     |
> |                    | halfcheetah | 93.02±0.40   | 90.81+-2.10  | **85.59+-5.09**  |
> |                    | walker2d    | 107.98±0.20  | 107.91+-0.16 | 107.53+-0.11     |
> |                    | ant         | 128.19±1.52  | 127.26+-2.48 | 127.30+-1.37     |
> | random+few-expert  | hopper      | 107.28±3.92  | 108.99+-0.76 | 109.34+-1.62     |
> |                    | halfcheetah | 88.77±1.26   | 86.62+-1.98  | **80.49+-6.41**  |
> |                    | walker2d    | 108.40±0.21  | 108.13+-0.29 | 107.78+-0.22     |
> |                    | ant         | 120.09±5.17  | 119.63+-2.40 | **105.98+-7.03** |
> | medium+expert      | hopper      | 110.20±0.51  | 109.89+-0.22 | 109.58+-1.18     |
> |                    | halfcheetah | 93.12±0.32   | 91.07±1.72   | **76.30+-13.56** |
> |                    | walker2d    | 108.12±0.13  | 108.31+-0.21 | 108.03+-0.15     |
> |                    | ant         | 124.72±3.49  | 127.71+-2.72 | 128.61+-0.71     |
> | medium+few-expert  | hopper      | 108.96±1.33  | 109.68+-0.56 | 108.20+-0.83     |
> |                    | halfcheetah | 89.47±0.82   | 90.20±1.94   | **77.65+-11.48** |
> |                    | walker2d    | 108.15±0.43  | 107.88+-0.83 | 107.39+-0.92     |
> |                    | ant         | 120.36±1.25  | 118.23+-5.04 | **102.14+-7.26** |
>
> > 3.  Have the authors conducted any experiments in image-based environments? Such settings may better reflect the nature of imitation from observation scenarios, where actions are not available and only visual state information is provided.
>
> We thank the reviewer for the insightful question. In this work, we focused on the well-established D4RL state-based benchmarks to allow for direct and rigorous comparison with recent state-of-the-art LfO methods (e.g., SMODICE, PW-DICE, DILO), which are also primarily evaluated in these settings.
>
> While we agree that extending to image-based tasks would broaden the applicability of our method, experiments on domains like V-D4RL [1] require careful design of visual encoders and significant effort to ensure fair integration with both our framework and existing baselines. As most prior LfO methods also do not evaluate in image-based settings, a thorough and fair comparison remains challenging.
>
> We will include a discussion in the revised paper highlighting the extension to vision-based tasks as a key direction for future work.
>
> **References:**
>
> [1]: Lu, Cong, et al. "Challenges and opportunities in offline reinforcement learning from visual observations." arXiv preprint arXiv:2206.04779 (2022).
>
>
> ---
>
> **We hope our revisions and our response  address the reviewers’ concerns and further clarify our contributions. If there are any additional questions or comments, we would be happy to address them.**

---

> > ### Comment · Reviewer_ahUM · 2025-08-03
> >
> > We thank the authors for the detailed response. I believe the authors have addressed the concerns raised in the review. I do not have other questions. I will also check all the other reviews and responses and decide whether to change the ratings or not.

---

> ### Author Response · Authors · 2025-08-08
> **Thank you for the feedback!**
>
> We sincerely thank the reviewer for carefully reading our paper, providing thoughtful responses, and offering valuable feedback. We will certainly incorporate the corresponding discussions and the additional experiments to the final version of the paper.
>
> We remain fully available and would be glad to provide any further details or clarify any remaining points. We believe our responses have addressed most of the concerns raised by you and the other reviewers.
>
> If our replies and the planned revisions have satisfactorily resolved your concerns, we would be grateful if you could kindly reflect this in your evaluation of our work.
>
> Thank you once again for your constructive engagement.

---

### Decision · Program_Chairs · 2025-09-17

**Decision:**

Accept (poster)

**Comment:**

(a) Summary of Claims and Findings

This paper introduces IOSTOM, a novel algorithm for offline Learning from Observations (LfO). IOSTOM learns expert policies from state-only expert data and a separate suboptimal state-action dataset, avoiding the common need for discriminators or inverse dynamics models. The authors propose a new objective based on matching the joint state transition occupancy between the learned policy and the expert. This is formulated as an unconstrained Q-learning problem, and following weighted behavior cloning to recover the implicit policy. Empirical results on D4RL benchmarks and a real-world navigation task show that IOSTOM significantly outperforms state-of-the-art methods.

(b) Strengths

The reviewers agree on several key strengths of the paper:
- Novelty: The action-free objective based on state transition occupancy is a novel and valuable contribution to the LfO field.
- Theoretical Soundness: The method is well-grounded in theory, with clear mathematical derivations that connect Q-learning and the implicit policy recovery.
- Strong Empirical Performance: The paper presents extensive and convincing results, with IOSTOM consistently outperforming baselines on diverse and challenging tasks.

(c) Weaknesses

Initial weaknesses raised by the reviewers included:
- Clarity: One reviewer (ahUM) initially find the paper's presentation dense and difficult to follow, citing missing definitions and unclear derivations.
- Experimental Gaps: Reviewers question the sensitivity to certain hyperparameters (y8cD, 4HqZ), the lack of comparison to a key baseline LobsDICE (y8cD), and the realism of the data requirements (gGMk, 4HqZ). Moreover, one reviewer (y8cD) raised concerns about the fairness of the offline evaluation.

(d) Reasons for Recommendation

I recommend accepting this paper. It presents a high-quality, technically solid contribution to the important problem of offline LfO. The proposed method, IOSTOM, is novel, theoretically well-motivated, and demonstrates superior performance.

The authors' rebuttal has addressed nearly all concerns raised by the reviewers. The initial clarity issues are not significant and the authors have committed to revisions to improve the paper's readability. The authors also provide substantial new experiments, including a full comparison against the requested LobsDICE baseline and an extensive ablation study on the key hyperparameter α. While one reviewer remains concerning about the hyperparameter sensitivity, the overall consensus on the paper's strengths—its novelty, theoretical rigor, and state-of-the-art results—outweigh the remaining concerns.

(e) Summary of Discussion and Rebuttal

The discussion period is very productive. The authors provide detailed responses and new experimental results that addressed the reviewers' main concerns.
- Clarity: In response to Reviewer ahUM, the authors agreed to clarify all missing definitions and derivations. This satisfied the reviewer, who subsequently raised their score.
- Hyperparameter Sensitivity: In response to Reviewers y8cD and 4HqZ, the authors provided a new, comprehensive ablation study across 24 tasks for the hyperparameter α. The results convincingly showed that the method is not overly sensitive to this parameter and that the default value is a robust choice.
- Comparison to LobsDICE: As requested by Reviewer y8cD, the authors ran a full suite of experiments comparing IOSTOM to LobsDICE. The results clearly demonstrated IOSTOM's superior performance.
- Offline Evaluation: Reviewer y8cD questioned the use of environment access for hyperparameter tuning. The authors clarified that this is a standard and widely accepted practice in the field, as fully offline model selection remains an open problem, and that their tuning was minimal and fairly conducted.

Following the rebuttal, three of the four reviewers lean toward acceptance, with two raising their scores. While Reviewer y8cD's final rating remains a borderline reject due to lingering concerns about the hyperparameter sensitivity, their primary technical and experimental questions are successfully addressed. The productive discussion and the authors' thorough response have further solidified that this submission is ready for acceptance.